# Feasibility study of multipixel retrieval of optical thickness and droplet effective radius of inhomogeneous clouds using deep learning

Rintaro Okamura[1], Hironobu Iwabuchi[1], and K. Sebastian Schmidt[2]

[1]Center for Atmospheric and Oceanic Studies, Graduate School of Science, Tohoku University, 6-3 Aoba Aramaki-aza, Sendai, Miyagi 980-8578, Japan
[2]Department of Atmospheric and Oceanic Sciences, University of Colorado, Boulder, CO, USA

*Correspondence to:* Hironobu Iwabuchi (hiroiwa@m.tohoku.ac.jp)

**Abstract.** Three-dimensional (3D) radiative transfer effects are a major source of retrieval errors in satellite-based optical remote sensing of clouds. The challenge is that 3D effects manifest themselves across multiple satellite pixels, which traditional single-pixel approaches cannot capture. In this study, we present two multi-pixel retrieval approaches based on deep learning, a technique that is becoming increasingly successful for complex problems in engineering and other areas. Specifically, we use deep neural networks (DNNs) to obtain multipixel estimates of cloud optical thickness and column-mean cloud droplet effective radius from multispectral, multi-pixel radiances. The first DNN method corrects traditional bi-spectral retrievals based on the plane-parallel homogeneous cloud assumption using the reflectances at the same two wavelengths. The other DNN method uses so-called convolutional layers and retrieves cloud properties directly from the reflectances at four wavelengths. The DNN methods are trained and tested on cloud fields from large-eddy simulations used as input to a 3D radiative transfer model to simulate upward radiances. The second DNN-based retrieval, sidestepping the bi-spectral retrieval step through convolutional layers, is shown to be more accurate. It reduces 3D radiative transfer effects that would otherwise affect the radiance values and estimates cloud properties robustly even for optically thick clouds.

## 1 Introduction

Clouds play an important role in determining the radiation budget of the Earth. To understand how, it is necessary to know the global distribution of cloud properties such as optical thickness (COT) and cloud droplet effective radius (CDER). These particular cloud properties are retrieved globally by optical remote sensing from various satellites. A standard method for COT and CDER retrieval is the bi-spectral method that is used to produce the Moderate Resolution Imaging Spectroradiometer (MODIS) cloud product (Nakajima and King, 1990; Platnick et al., 2003). This method uses solar reflection measurements at two wavelengths, one with and the other without absorption by water droplets. The nonabsorbing wavelength is selected in the visible or near-infrared part of the spectrum, whereas the absorbing one is in the shortwave infrared (SWIR) part, typically around 1.6, 2.1, or 3.7 $\mu$m. The method is based on the independent pixel approximation (IPA) assuming plane-parallel, horizontally and vertically homogeneous cloud for each pixel of the satellite image because of high computational

cost for simulation of three-dimensional (3D) radiative transfer. The observed cloud radiances result from three-dimensional (3D) radiative transfer in the cloud field, which means that the radiances are not only affected by the vertical cloud structure, but also by net horizontal radiative transport in inhomogeneous cloud fields. Previous studies have pointed out that cloud inhomogeneities and 3D radiative effects produce large errors in the retrieved cloud properties (Iwabuchi and Hayasaka, 2002, 2003; Zhang and Platnick, 2011; Zhang et al., 2012). Studies using observational data have confirmed the dependency of such retrieval errors on both the cloud horizontal and vertical inhomogeneity and the sun–cloud–satellite viewing geometry (Liang et al., 2009; Liang and Girolamo, 2013; Grosvenor and Wood, 2014).

Satellite image data with relatively coarse resolution does not contain sufficient information about in-pixel inhomogeneity. Although statistical bias correction is possible (Iwabuchi and Hayasaka, 2002), it is still difficult to perform error correction on each pixel, especially if unresolved in-pixel inhomogeneity is the major source of error. Zhang et al. (2016) recently described a novel method to correct the effect of in-pixel cloud inhomogeneity using subpixel reflectance variability. For finer-resolution imagery, by contrast, retrieval errors from inter-pixel horizontal radiative transport become more important. The radiance observed at each pixel is determined by the spatial arrangement of cloud water in the pixel of interest (target pixel) and its neighbors. The 3D radiative effects operate on horizontal scales that are determined mainly by cloud thickness and solar zenith angle. When the sun is oblique (i.e., with a solar zenith angle of $60°$ or larger), the maximum horizontal scale for 3D radiative effects is roughly 15–20 times larger than cloud thickness (Marshak and Davis, 2005). This necessitates consideration of the adjacent cloud effects when estimating the cloud properties at the target pixel. Iwabuchi and Hayasaka (2003) attempted to correct the horizontal transport effect by using multispectral, multipixel radiances for retrieving COT and CDER. They fitted a polynomial function of the multispectral radiances at the target and adjacent pixels to the IPA radiances at the target pixel. Since 3D radiative effects vary with COT, CDER, cloud geometrical thickness, cloud top roughness, and sun–cloud–satellite geometry, Iwabuchi and Hayasaka (2003) had to construct different sets of fitting coefficients, thus limiting the applicability of the technique in practice. In addition, their method was based on linear regression, which is not flexible to account for any nonlinear 3D radiative transfer effects.

To consider adjacency effects in a generalized manner, neural networks (NNs) (also known as multilayer perceptrons) are useful, and they have thus been applied to cloud detection and retrieval. Minnis et al. (2016) used an NN recently to estimate the COT of ice clouds from MODIS multispectral infrared radiances. An NN is expected to improve cloud retrieval accuracy in presence of 3D radiative effects because of the complexity of the problem. Therefore, a few studies have already proposed such applications to the problem of 3D clouds. Faure et al. (2001) demonstrated the feasibility of NNs to retrieve mean optical thickness, mean effective radius, fractional cloud cover, and subpixel-scale cloud inhomogeneity from multispectral radiance data at wavelengths of 0.64, 1.6, 2.2, and 3.7 $\mu$m for a pixel resolution of 0.8 km × 0.8 km. Faure et al. (2002) improved NN cloud property retrievals of 1D inhomogeneous clouds by considering multi-spectral radiance (at 0.64, 1.6, 2.2, and 3.7 $\mu$m ) from a collection of pixels adjacent to the pixel of interest. Cornet et al. (2004) used NNs to retrieve cloud properties (i.e., mean optical thickness, mean effective radius, fractional cloud cover, inhomogeneity parameters of optical thickness and effective radius, and cloud-top temperature) from multispectral and multiscale radiance data. They used horizontal resolutions of 0.25 km × 0.25 km at wavelengths of 0.544, 1.6, and 2.15 $\mu$m and 1 km × 1 km at wavelengths of 0.544, 1.6, 2.15, 3.65,

and 10.8 $\mu$m. Their method was adapted to MODIS in Cornet et al. (2005). Evans et al. (2008) used an NN to estimate mean and standard deviations of COT from multiangular reflectances of boundary layer clouds.

More recently, deep learning (a machine-learning technique), which uses deep neural networks (DNNs), has become a useful tool in various applications. Deep learning involves training a DNN that has three or more layers with a network structure that is more complex than that used previously. An advantage of deep learning is automatic feature extraction: features in training datasets are learned hierarchically in the DNNs, although it is not easy to trace how the features emerge. Nevertheless, DNNs extend the applicability of NN to more complex problems. In addition, input and output parameters can easily be added, and structures can be modified - another advantage of deep learning. In combination, these advantages make DNNs more flexible under varying conditions. They are more suitable for approximating complex nonlinear functions of many variables because the degree of nonlinearity increases with the number of layers, and the ability to approximate a function generally improves for a deeper NN. Recent advances in computer technology, such as multicore central processing units (CPUs) and general-purpose graphics processing units (GPGPUs), have facilitated calculations involving the large training datasets that are required for DNNs. In addition, a number of DNN optimization algorithms have been proposed in the past few years.

The present study is aimed at using a DNN approach to retrieve the COT and CDER of inhomogeneous clouds, and at testing the feasibility of a multispectral, multipixel approach based on DNNs. For training and testing, we use 3D cloud-field data generated by large-eddy simulation (LES) and radiances generated by a 3D radiative transfer model. The outline of this paper is as follows. Section 2 explains the cloud-field data and radiative-transfer simulations that are used to generate the training and test datasets. Section 3 describes the designs and configurations of our DNNs and the preprocessing methods. Section 4 presents results of performance comparisons for cloud retrieval using DNNs, IPA-based look-up table (LUT), and a simple NN. Finally, Section 5 concludes the paper with a discussion on the merits of DNN-based cloud retrieval.

## 2 Data

### 2.1 SCALE-LES cloud-field data

Three-dimensional cloud-field data are generated using an LES model known as SCALE-LES (Sato et al., 2014, 2015; Nishizawa et al., 2015). A double-moment bulk scheme is used for the cloud microphysics. The cloud liquid-water mass mixing ratio and number density are obtained at each grid point in the domain. Figure 1 shows examples of such cloud-field data for two types of boundary-layer cloud: closed-cell and open-cell. These cloud types are simulated for polluted (closed) and clean (open) aerosol conditions (Sato et al., 2014). Clouds are optically thick in the closed case, whereas they are optically thin with large precipitation rates in the open case. Each case consists of 60 time steps at 1-min intervals. The horizontal size of the LES scenes is 28 km $\times$ 28 km. The resolution for the x- and y-axis of the cloud field is originally 35 m. For the z-axis, the resolution is 5 m at the bottom of the atmosphere, and it is coarse (less than 60 m) for the upper layers. Area averaging was done over a cloud region of 280 m for x- and y-axis. For simplicity, subpixel clouds are not considered in this study. After the

averaging, the finest resolution for z-axis is 40 m. The CDER is calculated as

$$r_e = \frac{1}{\chi} \left( \frac{3}{4\pi} \frac{\mathrm{LWC}}{\rho_b N} \right)^{\frac{1}{3}},$$ (1)

where $\chi$ is a constant depending on width of the droplet size distribution, LWC is the liquid water content, $\rho_b$ is the density of water, and $N$ is the droplet number density.

As shown in Fig. 1, the extinction coefficient and CDER in both cases tend to increase with height from the cloud base toward the cloud top, although the IPA retrieval assumes a homogeneous cloud. The CDER has a particularly inhomogeneous vertical structure in the closed-cell case. In the open-cell case, the CDER spatial variability is high in general, particularly so in the uppermost core parts of cells. There is no community consensus on a single definition of CDER that is representative of the full column in the case of a vertically inhomogeneous cloud. Nevertheless, this study introduces the retrieval of such a representative CDER, where the vertically-averaged LWC and number density are used to define the column-mean effective radius $R_e$:

$$R_e = \frac{1}{\chi} \left( \frac{3}{4\pi} \frac{\langle \mathrm{LWC} \rangle}{\rho_b \langle N \rangle} \right)^{\frac{1}{3}},$$ (2)

where $\langle . \rangle$ denotes the mean over cloud column. Note the similarity between the definition of $R_e$ in Eq. (2) and that of $r_e$ in Eq. (1). $R_e$ represents droplet size of a cloud column. The retrieval performance for this parameter will be discussed in Section 4. It should be pointed out that there are other possibilities for column-average CDER (Miller et al., 2016).

Figure 2 shows temporal variations of (a) the domain-mean COT (specified at 0.55 $\mu$m throughout this paper), (b) the domain-mean column-mean CDER, (c) the cloud fraction, and (d) the inhomogeneity index $H$, defined as

$$H = \frac{\sigma_\tau^2}{\overline{\tau}^2},$$ (3)

where $\sigma_\tau$ is the standard deviation of the COT and $\overline{\tau}$ is the mean COT. The coefficient of COT variation, $\sqrt{H}$, has been used often in previous studies (Szczap et al., 2000; Liang et al., 2009; Liang and Girolamo, 2013). Clouds in the closed-cell case are optically thick and horizontally homogeneous, covering almost the entire sky and giving a high cloud fraction. Therefore, as can be seen in Fig. 2(a, b), the domain-averaged COT and CDER remain almost constant over the entire period. In contrast, clouds in the open-cell case are distributed sparsely, meaning that the inhomogeneity index $H$ is larger than that in the closed-cell case and increases gradually over time. The domain-averaged CDER is larger in the open-cell case than it is in the closed-cell case.

## 2.2 Radiative transfer simulations

A Monte Carlo 3D radiative transfer model known as MCARaTS (Monte Carlo Radiative Transfer Simulator; Iwabuchi (2006)) is used to simulate the cloud radiances. The radiances reflected in the zenith direction are calculated for solar zenith angles (SZAs) of $20°$ and $60°$ at wavelengths of 0.86, 1.64, 2.13, and 3.75 $\mu$m. The aerosol optical properties are derived using the one-dimensional RSTAR6b radiative transfer code (Nakajima and Tanaka, 1986, 1988). The aerosol optical thickness is assumed to be 0.2, and the rural aerosol model is used (Hänel, 1976). A correlated $k$-distribution is used for gaseous absorption

by $H_2O$, $CO_2$, $O_3$, $N_2O$, $CO$, $CH_4$, and $O_2$ molecules (Sekiguchi and Nakajima, 2008). Rayleigh scattering by air molecules is included in the scattering process. The particle size distribution of water cloud droplets is expressed as a log-normal *volume* ($V$) distribution

$$\frac{\mathrm{d}V}{\mathrm{d}\ln r} = C \exp\left[-\frac{1}{2}\left(\frac{\ln r - \ln r_{\mathrm{mod}}}{\ln s}\right)^2\right], \tag{4}$$

where $r$ is the particle radius, $C$ is the maximum value of the volume distribution at mode radius $r_{\mathrm{mod}}$, and $s$ is the width of the distribution. In this study, we assume $s = 1.5$. The CDER $r_{\mathrm{e}}$ is related to $r_{\mathrm{mod}}$ by $r_{\mathrm{e}} = r_{\mathrm{mod}} \exp(-1/2 \times (\ln s)^2)$. The $\chi$ parameter in Eqs. (1) and (2) is determined as $\chi = r_{\mathrm{vol}}/r_e = \exp(-\ln^2 s) = 0.84$, where $r_{\mathrm{vol}}$ is the volume mean radius. The scattering properties of water cloud droplets are calculated using the Lorenz–Mie theory (Bohren and Huffman, 1983). For simplicity, the underlying surface is approximated as black.

## 3 Method

### 3.1 Deep learning techniques

This section briefly explains fundamentals of deep learning. This study uses Chainer, an NN framework developed by Tokui et al. (2015), for implementating DNN cloud retrieval methods. Chainer is used in a wide variety of research fields because it covers common functions and algorithms for constructing and training DNNs and provides easy access to efficient GPU-based computation. It is easy to test different DNN structures and learning techniques, and there is high degree of freedom for choice. Interested readers may refer to Tokui et al. (2015) for details. Several other deep learning frameworks have been developed for general purposes and are publicly available.

The DNNs consists of multiple network layers, each of which consists of units. Each unit receives input signals from the previous layer and generates an output signal. If each unit in a layer is connected with all units in the previous layer, that layer is called fully connected. The unit computes a weighted sum and adds a bias as follows:

$$x = \sum_k w_k x'_k + b, \tag{5}$$

where $x'_k$ is the $k$th input signal, $w_k$ is the corresponding weight, and $b$ is the bias. The weights and the bias are determined at the training stage. The result $x$ is usually transformed by a function known as the activation function to obtain an output signal. In this study, we use a rectified linear function (Nair and Hinton, 2010) defined as

$$f(x) = \max(0, x) \tag{6}$$

for the activation function. Among the various activation functions used for NNs, this rectified linear function is relatively simple, leads to good learning efficiency, and is most commonly used in recent DNN applications.

Image recognition is often implemented through convolutional NNs because they enable pattern recognition in an image. A convolutional layer consists of units that compute the convolution on the input image. For multichannel images, multiple structural filters operate within a convolutional layer. A convolutional signal $x_m$ of the $m$th output channel at a pixel is represented

as

$$x_m = \sum_l \sum_k w_{k,l,m} x'_{k,l,m} + b_m, \qquad (7)$$

where $x'_{k,l,m}$ and $w_{k,l,m}$ are the input signal and the corresponding filter weight for the $k$th pixel (the target or an adjacent pixel) and $l$th input channel. The number of required filters is the product of the number of input channels (wavelengths) and output channels. The summation over $k$ in Eq. (7) operates only on the target pixel and its neighbors. Unlike a fully connected layer, a convolutional layer has the following two characteristics: 1) the input and output signals of a convolutional layer are sparsely connected, and 2) the filter profiles are defined independently for input channels but are shared among all pixels; the filter profile does not depend on pixel location in the input image.

Current deep learning techniques are flexible to use a DNN structure by connecting multiple fully-connected and convolutional layers in a complicated way, as shown in the next subsection. During the training, the DNN parameters are optimized to minimize the so-called loss function. In this study, the loss function is the sum of the squared residuals between the DNN output and ideal data in the training dataset. For the DNN optimization, we use the Adam (Adaptive moment estimation; Kingma and Ba (2014)) algorithm, which automatically determines the learning rate at each training step using the mean and variance of the loss function. An NN is expected to deliver meaningful and accurate retrievals for the dataset that it was trained on. However, in some cases, the NN can be overfitted to the training dataset, thereby losing its ability to generalize, and performing appreciably worse for other data. Such overfitting is a serious issue in NNs. In the present study, we use the dropout technique (Srivastava et al., 2014) to overcome this problem. The dropout technique removes randomly selected units from the NN at each step in the training stage, decreasing the number of degrees of freedom of the NN, thus avoiding overfitting. An NN trained with the dropout technique can work like ensemble estimation that uses many different independently trained NNs. Dropout results in better performance and is widely used in many applications.

### 3.2 Design and configuration of DNNs

The DNNs used in this study are designed to estimate COT and column-mean CDER simultaneously at multiple pixels from multipixel, multispectral radiances. This is a unique approach compared to previous studies. Larger input and output vectors allow more degrees of freedom for the features to be learned in the DNNs. Two types of DNN were constructed:

1. DNN-2r (with IPA retrieval and two wavelengths) that corrects IPA retrievals based on 0.86 and 2.13 $\mu$m radiances using the radiances at those same wavelengths (0.86 and 2.13 $\mu$m);

2. DNN-4w (with four wavelengths) that uses convolutional layers and retrieves cloud properties directly from the radiances at 0.86, 1.64, 2.13, and 3.75 $\mu$m.

It should be noted that only solar radiation is considered in the present study, which requires that the thermal radiation at 3.75 $\mu$m be corrected during pre-processing.

The DNN-2r network is designed to correct the IPA retrieval of COT and CDER from the bispectral retrieval for 3D effects. The elements of the DNN-2r input vector are the radiances at the wavelengths of 0.86 and 2.13 $\mu$m. Prior to applying DNN-

2r, the COT and CDER are estimated by IPA for $10 \times 10$ pixels at 280 m resolution. Thus, the input vector has $400 = (10 \times 10 \times (2+2))$ elements. Figure 3 shows the DNN-2r structure schematically; the COT and CDER distributions are estimated at $8 \times 8$ pixels at the center of the input field, and the output vector has $128 = (8 \times 8 \times 2)$ elements. The reason for including margin pixels in the input field is to take into account the 3D radiative effects from the surroundings of the cloud field. The DNN-2r network consists of several fully connected layers. In the first layers, radiances and IPA-estimated cloud properties are merged to obtain $8 \times 8 \times 2$ elements (two elements per pixel for $8 \times 8$ pixels). The final part of DNN-2r consists of two independent groups of layers that finally estimate the COT and CDER. As in the residual network designed by He et al. (2015), the DNN-2r network has what are known as shortcuts, which allow residuals to be learned. The NN is trained to predict the correction terms to be added to the data from the shortcut path represented in Fig. 3 as bypassing route of data. Such shortcuts make machine learning possible even in cases with many NN layers. In this way, the DNN-2r network can be considered a way to correct the IPA retrievals.

The DNN-4w structure is shown schematically in Fig. 4. The input comprises radiance distributions at four wavelengths (0.86, 1.64, 2.13, and 3.75 $\mu$m) and $10 \times 10$ pixels of 280 m resolution. Thus, the input vector has $400 = (10 \times 10 \times 4)$ elements. Unlike in DNN-2r, the COT and CDER distributions in DNN-4w are predicted at the center of $6 \times 6$ pixels of the input field, and the output vector has $72 = (6 \times 6 \times 2)$ elements. In addition to shortcuts, the DNN-4w network has two convolutional layers. In the first convolutional layer, convolutions operate on $5 \times 5$ pixels surrounding the center pixel, with 100 different profiles of filter weights for each wavelength. There are 400 filters in the first convolutional layer because the number of input wavelengths is 4 and that of output channels is 100. As shown in Fig. 4, the activation function is applied to the signal $x_m$ in the first convolutional layer, but not in the second. By using these convolutional layers, we expect that the DNN learns image patterns that capture the 3D radiative effects between the target pixel and its surroundings. The DNN-4w network firstly corrects for 3D radiative transfer effects and then transforms the signals to COT and CDER with the possibility of additional 3D corrections at this stage.

The above two DNN structures were obtained from various trial-and-error experiments. Different DNN structures were also tested. For example, we tested a DNN similar to DNN-2r but with four wavelengths, and one similar to DNN-4w but with only two wavelengths. However, DNN-2r and DNN-4w performed best. There is room for improvement in DNN structures, which should be investigated in the future.

### 3.3 Generation of the training and test datasets

A training dataset is necessary for machine learning. In this study, the training dataset is generated as follows. The zenith radiances are calculated using MCARaTS with $10^5$ model photons incident on each pixel, which results in Monte Carlo noise of approximately $1\%$. Such noise can be interpreted as measurement noise in the present problem. From two cases of SCALE-LES cloud-field data, $1,977,440$ samples ($10 \times 10$ areas) are chosen randomly for the training datasets. As shown in Fig. 2, the 25th to 75th percentile ranges for COT are 0–5 and 11–15 for the open- and closed-cell cases, respectively. With a DNN, a variety of training data is important for better generalization performance. To increase the variety of the COT training data, one half was generated from original cloud data, whereas the other half was generated from artificially modified cloud fields

in which the cloud extinction coefficients were multiplied by numbers chosen randomly from the range 0.5–1.5. The cloud extinction coefficients of all pixels within a single cloud scene were multiplied by the same number.

Although DNN can generally approximate nonlinear functions, it is expected that less nonlinear functions can be approximated by fewer DNN layers. Constructing efficient DNN thus makes it desirable to linearize the relationship between input and output variables to some degree. Because the radiances are highly nonlinear with respect to the COT and CDER, it is convenient to transform the COT and CDER by some simple functions. In the DNN preprocessing, the cloud properties are transformed using

$$F(\tau) = \frac{(1-g)\tau}{1+(1-g)\tau}, \tag{8}$$

$$G(r_e) = \sqrt{r_e}, \tag{9}$$

where $g$ is the asymmetry parameter. As a representative value for water droplets, we set $g = 0.86$ for preprocessing purposes only. After the above transformations, all the DNN input and output data, including the radiances and cloud properties, are normalized as

$$\boldsymbol{z}'_{i,j} = \frac{\boldsymbol{z}_{i,j} - \overline{z_j}}{\sigma_j}, \tag{10}$$

where $\boldsymbol{z}_{i,j}$ is the $j$th element of an input or output vector in the $i$th sample, and $\overline{z_j}$ and $\sigma_j$ are the mean and standard deviation, respectively, of the $j$th element over the all samples. This is referred to as $z$-score normalization and is known to improve the efficiency of a DNN (Kotsiantis et al., 2006; Nawi et al., 2013).

The test dataset used for evaluation should be independent of the training dataset. In the present study, the training and test datasets include different randomly selected locations within the cloud fields, but the statistics of cloud properties are nearly identical in the training and test datasets. The test datasets include $10,000$ samples. As for computational cost, the training requires significant computation time, for which even one GPU helps considerably. Once the DNNs are trained, the retrievals using the present DNNs are generally very quick because they entail only very few simple manipulations of numerical data.

## 4 Results

In this section, we illustrate the ability of DNNs to retrieve cloud properties, and we compare it with the corresponding abilities of existing methods. The values of COT and CDER are retrieved from test datasets by using DNNs and IPA retrievals. The retrieved values are compared to the true values in the test datasets, and the retrieval errors at each pixel are evaluated. In the IPA retrieval, COT and CDER are estimated from LUTs of radiances at the wavelengths of 0.86 and 2.13 $\mu$m. These wavelengths are used in the MODIS product for retrieving cloud properties over oceans (Platnick et al., 2003). The lower and upper limits for COT are zero and 150, respectively, and those for CDER are zero and 55 $\mu$m, respectively. If any radiance falls outside the associated range defined by the LUTs, the COT/CDER value is forced to be the lower or upper limit, as appropriate.

## 4.1 Retrieval results for DNN-2r and DNN-4w

Figure 5 shows examples of the IPA and DNN-4w retrieval results for an open-cell case with a SZA of $60°$ and a viewing zenith angle of $0°$. Cross sections at $y = 14.56$ km are shown in Fig. 6 with additional DNN-2r retrieval results. In Fig. 6, the relative error of estimated cloud properties are also shown. On this cross section, the RMSE of estimated cloud properties are shown in Table 1. The sunny (left-hand) side of the COT fluctuation peak is directly illuminated by the Sun. This is noticeable at locations around 11 km and 16.5 km. For pixels on that side, the radiances calculated by 3D radiative transfer are brightened (illumination effect), which results in an overestimation (underestimation) of COT (CDER) by IPA retrievals. For pixels on the opposite (right-hand) side, the radiances are darkened (shadowing effect), and COT (CDER) is underestimated (overestimated) by IPA. These illumination and shadowing effects have considerable influence on the IPA retrieval (Várnai and Davies, 1999; Várnai and Marshak, 2002; Marshak et al., 2006). These effects lead to a spatial offset in the IPA-retrieved horizontal COT distribution, and the COT IPA retrieval error is particularly large for optically thick pixels. This can be ascribed to weaker sensitivity of radiance to COT when COT is larger; a small difference in radiance leads to large difference in retrieved COT. Compared to the IPA retrieval, the DNNs retrieve COT values that are closer to the true values assumed in this test. The DNN-4w successfully corrects the phase lag as shown in Fig. 6c. However, minor errors are still present in the DNN-retrieved COT.

As for the retrieved values of CDER, the DNN ones are obviously better than the IPA-retrieved ones. The CDER is noticeably overestimated at pixels for which the shadowing effect decreases the radiance, a fact that we attribute to the strong nonlinear dependence of CDER on radiance. As a result, a positive bias appears in the IPA-retrieved CDER, which also shows an appreciable fluctuation at small horizontal scales because SWIR radiances are sensitive to cloud-top variability at such scales (Iwabuchi and Hayasaka, 2003). In contrast, the DNN-retrieved CDER is generally highly accurate, although small-scale fluctuations of CDER are not very well reproduced.

The COT and CDER retrieval errors are evaluated for all the test datasets, and the mean and standard deviation of the relative errors are calculated in bins that are equally spaced in the logarithm of COT and CDER. The results are evaluated using $360,000$ pixels for each SZA. In Fig. 7, the IPA retrieval and DNN-4w relative errors are plotted against the true COT and CDER values. The IPA-retrieved COT error and its standard deviation are particularly large for a SZA of $60°$, where radiative roughening causes the 3D radiance to deviate from the IPA radiance. Both the COT and CDER retrieval errors are reduced considerably by using the DNN, which suggests that the DNN is well trained to correct the 3D radiative transfer effects. The DNN mean bias errors are generally closer to zero than those from the IPA retrieval. Compared to the IPA retrieval, the DNN retrieves COT better, even at optically very thick pixels. In particular, the COT error is markedly reduced for true COT values greater than 5 and for an SZA of $60°$. At pixels with small COT (1 or less), the DNN overestimates COT, although the errors are still smaller than those from the IPA retrieval.

The DNN also yields better CDER retrievals than does the IPA retrieval, with much smaller variability of CDER errors. For SZAs of $20°$ and $60°$, the IPA-retrieved CDER tends to be overestimated over almost the entire range of CDER. The IPA retrieval shows a particularly large bias when the true CDER is small, although very few data are available for CDER

values less than 15 $\mu$m, as shown in Fig. 2. This overestimation of CDER can be partly attributed to the neglect of vertical inhomogeneity in the IPA retrieval. The reflected SWIR radiances (2.13 $\mu$m) give information about the cloud microphysical status only near the cloud top (Platnick, 2000), and the IPA-retrieved CDER is associated primarily with the CDER near the cloud top (Nakajima et al., 2010; Zhang et al., 2012; Nagao et al., 2013). IPA-retrieved CDER thus tends to be larger than column-mean CDER, whereas DNNs are by design trained to learn the relationship between the column-mean CDER and radiances by taking into account the vertical inhomogeneity. However, this vertical inhomogeneity effect on the IPA-retrieved CDER does not seem to be the main cause of the large positive bias. Overestimation of CDER in the IPA retrieval is mainly observed at the shadowed pixels, as shown in Figs. 5 and 6. The IPA retrieval also shows large values of standard deviation of the relative errors, particularly for small values of CDER. Figures 5(a) and 5(b) show that the CDER tends to be smaller at pixels with small COT, where the shadowing tends to reduce the SWIR radiance. A small radiance perturbation due to 3D effects may result in a large error in the retrieved CDER because of the weaker sensitivity of SWIR radiance to CDER in cases of small COT. However, the DNN-retrieved values of column-mean CDER are close to the true values.

Table 2 shows the relative RMSE of estimated COT and CDER by the IPA and DNN retrievals for open-cell and closed-cell cases. In both cases, the retrieval accuracies for DNNs are obviously improved compared to the IPA retrieval. An exception appears for COT in closed-cell case when SZA is 20°; COT RMSE of 24% for DNN is larger than 16% for IPA. The DNN-4w is better than DNN-2r in both cases.

Figure 8 shows selected examples of the trained $(5 \times 5)$-pixel filters of the first convolutional layer used in DNN-4w for a SZA of 60°. It is noted that the convolutional layer is designed to correct the 3D radiative effects appeared in radiances (Fig. 4). Only 16 out of 100 filters are shown here, and each filter weight can be either positive or negative. At the beginning, filters are initialized with white noise. Once the DNN has been well-trained, this noise is replaced with features that emerge from patterns in the input images. The patterns in some filters (e.g., first and eleventh ones from the left) are nearly symmetrical around the center pixel with various spatial profiles, which suggests that they extract features that characterize the relationship between the center pixel and those adjacent to it. For example, isotropic smoothing and second-order central difference operators have such a symmetrical pattern. Also, several filters (e.g., fifth one from the left) have higher weights in pixels along the solar azimuth direction, which suggests a feature related to the direct beam that operates along that direction. In our design of DNN-4w, different filters operate at each wavelength independently, whereas most of the obtained filters show strong correlations among wavelengths. These patterns suggest that the combination of filter patterns in the DNN corrects 3D radiative effects and thus recovers the local cloud property information. However, it is presently difficult to understand which combinations of filter patterns perform such corrections in the DNN, or indeed how they do so.

## 4.2 Comparison with previous work using a neural network

It is of interest to compare the performance of our present DNN with that of the NN used previously as the second method in Faure et al. (2001). Originally, this NN had two hidden layers with 10 units each and it used the 0.8 km × 0.8 km area-averaged radiances at four wavelengths (0.64, 1.6, 2.2, 3.7 $\mu$m) at the target pixel and eight adjacent pixels (called as "with ancillary data"). It is described in the section 3.3 (2) in the original paper. However, in this comparison, we construct an NN with 512

units in each layer to allow more degrees of freedom. The NN inputs for the present study are the radiances at four wavelengths (0.86, 1.64, 2.13, and 3.75 $\mu$m) at the target pixel and eight adjacent pixels, as in the second method in Faure et al. (2001), and the outputs are COT and CDER. In this comparison, the resolution of the pixels is 280 m. Data used for training and test for the NN are from the same original datasets used for DNNs.

5    It is of interest to compare the performance of our present DNN with that of the NN used previously as the second method in Faure et al. (2001), section 3.3 (2). Originally, this NN had two hidden layers with 10 units each and it used 0.8 km $\times$ 0.8 km area-averaged radiances at four wavelengths (0.64, 1.6, 2.2, 3.7 $\mu$m) at the target pixel and eight adjacent pixels (denoted as "ancillary data"). Rather than reproducing their NN exactly, we construct an NN with 512 units in each layer for the sake of this comparison to allow more degrees of freedom. As NN inputs, we use the radiances at four wavelengths (0.86, 1.64, 2.13, and 3.75 $\mu$m) at the target pixel and eight adjacent pixels. As in Faure et al. (2001), the outputs are COT and CDER, but with a pixel resolution of 280 m. Data for training and testing the NN are from the same original datasets used for DNNs.

Figure 9 shows comparisons of the NN and our DNNs. For a SZA of $20°$, the COT is well retrieved for true COTs of 10–50 for both the NN and DNNs. When the true COT is less than 10, the COT values from the NN and DNN-4w retrievals are overestimated more for optically thinner clouds, although DNN-2r gives better estimates. The COT estimated by the NN tends to be underestimated when the true COT is larger than 50, whereas DNN-2r and DNN-4w yield better retrievals in this range. For an SZA of $60°$, the DNN retrievals of COT are generally better than the NN retrievals. The COT retrievals by the NN tend to be overestimated (resp. underestimated) for optically thin (resp. thick) clouds. This suggests that 3D radiative effects with low sun are not well modeled in the current NN because it uses only $3 \times 3$ pixels, whereas the DNNs use $10 \times 10$ pixels. Moreover, the multiple convolutional layers in the DNNs are more powerful for representing the complex 3D radiative effects compared to the layers in the NN. In general, the DNN-2r retrievals show large error variability, with the largest standard deviation among the three methods. The CDER is well retrieved by all three methods (NN and DNNs) when the true CDER is larger than 10 $\mu$m, although overestimating smaller CDERs is common among the three methods.

## 5    Conclusions

In this study, the feasibility of a multispectral, multipixel approach to retrieving COT and CDER with a deep learning technique has been investigated. Two types of DNN were constructed: 1) DNN-2r that corrects IPA retrievals using the reflectances at two wavelengths, and 2) DNN-4w that uses convolutional layers and retrieves cloud properties directly from the reflectances at four wavelengths. Both DNNs retrieve multipixel estimates of COT and CDER simultaneously from multispectral, multipixel radiances. The DNNs were trained and evaluated with SCALE-LES cloud-field data at a horizontal resolution of 280 m. Both DNNs outperformed IPA-based retrieval in terms of accuracy, and showed better ability to represent 3D radiative effects compared to that of an NN used in previous work. The CDER retrievals of both DNNs were considerably better than the corresponding IPA retrieval. Whereas the IPA retrieval appreciably overestimated the CDER at pixels that were affected by shadowing, the DNNs successfully corrected such 3D effects. The DNN-4w network was generally more accurate than the DNN-2r network. Information that was lost in the IPA retrieval when the radiances came from LUTs made for plane-parallel

clouds limited the ability of the DNN-2r network to correct those retrievals sufficiently well. In contrast, the DNN-4w network does not use an IPA retrieval in its input, and therefore is more robust at retrieving cloud properties. In addition, multipixel information and convolutional layers were shown to be efficient in improving cloud retrievals with 3D radiative effects taken into account.

In the DNN-4w that we tested, we excluded 3D radiative transfer effects that occurred at horizontal scales greater than 560 m (2 pixels). In addition, we only considered a cloud thickness of less than 0.9 km, as shown in Fig. 1. It would therefore be interesting to test the sensitivity and performance of the algorithm for input vectors for wider areas (more pixels) of cloud. This is because 3D radiative transfer effects are known to operate on horizontal scales that are determined mainly by cloud thickness and solar zenith angle (Marshak and Davis, 2005). In the future, the application of DNNs to cloud remote sensing is expected
to become more common. However, using DNNs with actual satellite data will require training using realistic cloud fields for various types of cloud. Incorporating more parameters (e.g., sun–cloud–satellite geometry, surface albedo, aerosols, spectral and spatial specifications of sensors) into the method will also be necessary to handle the complexities of such measurement data. It is an advantage of this method that it is easy to incorporate such parameters into DNN. An appropriate DNN architecture for addition of input parameters should be investigated in the future.

*Acknowledgements.* SCALE-LES was developed by the SCALE interdisciplinary team at the RIKEN Advanced Institute for Computational Science (AICS), Japan. Some of the results in this paper were obtained using the K supercomputer at RIKEN AICS. The authors are grateful to Dr. Yousuke Sato of RIKEN for providing the SCALE-LES simulation data. Some of the results of radiative transfer calculations in this paper were obtained using the server and visualization systems of the Graduate School of Simulation Studies, University of Hyogo. We would also like to thank the OpenCLASTR project for the use of the RSTAR radiative transfer model. This work was partly supported by
a Grant-in-Aid for Scientific Research (B) (KAKENHI Grant No. 15H03729) of the Japan Society for the Promotion of Science (JSPS). Sebastian Schmidt was supported through NASA grant NNX15AQ19G (Remote Sensing Theory).

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

**Table 1.** The RMSE of estimated COT and CDER by the IPA retrieval and DNNs for a view zenith angle of $0°$ at $y = 14.56$ km in Fig. 5.

| Retrieval method | COT | CDER [micron] |
|---|---|---|
| IPA retrieval | 11.36 | 9.33 |
| DNN-2r | 5.64 | 1.64 |
| DNN-4w | 4.40 | 1.37 |

Srivastava, N., Hinton, G. E., Krizhevsky, A., Sutskever, I., and Salakhutdinov, R.: Dropout: a simple way to prevent neural networks from overfitting., Journal of Machine Learning Research, 15, 1929–1958, 2014.

Szczap, F., Isaka, H., Saute, M., Guillemet, B., and Ioltukhovski, A.: Effective radiative properties of bounded cascade nonabsorbing clouds: Definition of the equivalent homogeneous cloud approximation, Journal of Geophysical Research: Atmospheres, 105, 20 617–20 633, 2000.

Tokui, S., Oono, K., Hido, S., and Clayton, J.: Chainer: a Next-Generation Open Source Framework for Deep Learning, in: Proceedings of Workshop on Machine Learning Systems (LearningSys) in The Twenty-ninth Annual Conference on Neural Information Processing Systems (NIPS), http://learningsys.org/papers/LearningSys_2015_paper_33.pdf, 2015.

Várnai, T. and Davies, R.: Effects of cloud heterogeneities on shortwave radiation: Comparison of cloud-top variability and internal heterogeneity, Journal of the Atmospheric Sciences, 56, 4206–4224, 1999.

Várnai, T. and Marshak, A.: Observations of three-dimensional radiative effects that influence MODIS cloud optical thickness retrievals, Journal of the Atmospheric Sciences, 59, 1607–1618, 2002.

Zhang, Z. and Platnick, S.: An assessment of differences between cloud effective particle radius retrievals for marine water clouds from three MODIS spectral bands, Journal of Geophysical Research: Atmospheres, 116, https://doi.org/10.1029/2011JD016216, 2011.

Zhang, Z., Ackerman, A. S., Feingold, G., Platnick, S., Pincus, R., and Xue, H.: Effects of cloud horizontal inhomogeneity and drizzle on remote sensing of cloud droplet effective radius: Case studies based on large-eddy simulations, Journal of Geophysical Research: Atmospheres, 117, https://doi.org/10.1029/2012JD017655, 2012.

Zhang, Z., Werner, F., Cho, H.-M., Wind, G., Platnick, S., Ackerman, A., Di Girolamo, L., Marshak, A., and Meyer, K.: A framework based on 2-D Taylor expansion for quantifying the impacts of subpixel reflectance variance and covariance on cloud optical thickness and effective radius retrievals based on the bispectral method, Journal of Geophysical Research: Atmospheres, 121, 7007–7025, 2016.

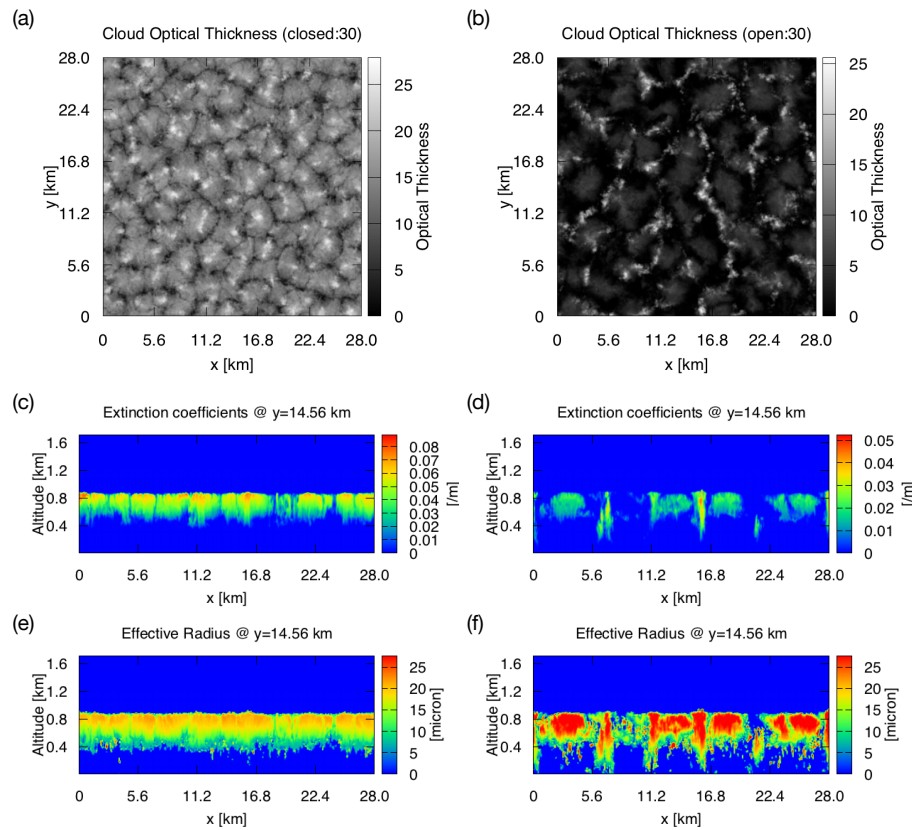

**Figure 1.** Examples of cloud properties in (a,c,e) closed-cell and (b,d,f) open-cell cases, taken from the 30th timestep of SCALE-LES simulation data. (a,b) Horizontal distributions of COT, (c,d) vertical cross sections of extinction coefficients, and (e,f) vertical cross sections of CDER.

**Table 2.** The relative RMSE of estimated COT and CDER by the IPA retrieval and DNNs for each cases of cloud field.

| Retrieval method | open-cell, COT | closed-cell, COT | open-cell, CDER | closed-cell, CDER |
|---|---|---|---|---|
| IPA retrieval, SZA: 20° | 30.7% | 16.0% | 38.3% | 51.2% |
| DNN-2r, SZA: 20° | 24.3% | 24.6% | 8.1% | 8.9% |
| DNN-4w, SZA: 20° | 21.6% | 23.4% | 5.5% | 6.7% |
| IPA retrieval, SZA: 60° | 74.8% | 50.9% | 51.1% | 55.2% |
| DNN-2r, SZA: 60° | 33.2% | 37.4% | 7.6% | 8.4% |
| DNN-4w, SZA: 60° | 26.6% | 18.7% | 6.5% | 7.3% |

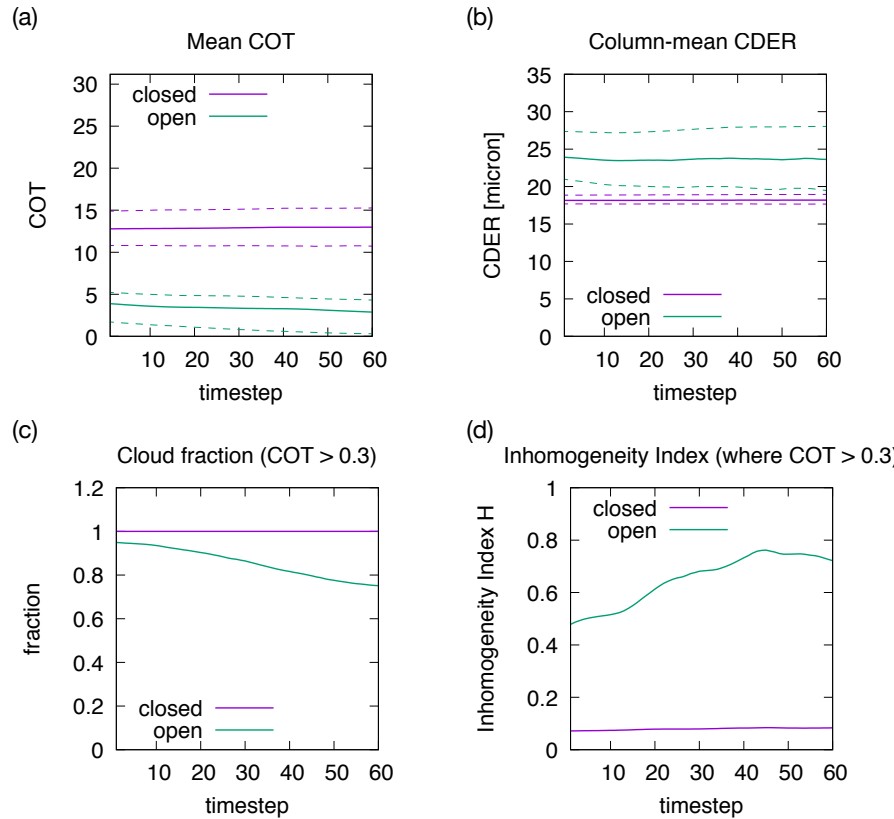

**Figure 2.** Temporal variations of (a) COT, (b) column-mean CDER, (c) cloud fraction, and (d) inhomogeneity index $H$. Solid lines shows mean values, and dashed lines show the 25th and 75th percentiles. The cloud fraction and $H$ are computed for pixels with COT > 0.3. A time step corresponds to one minute.

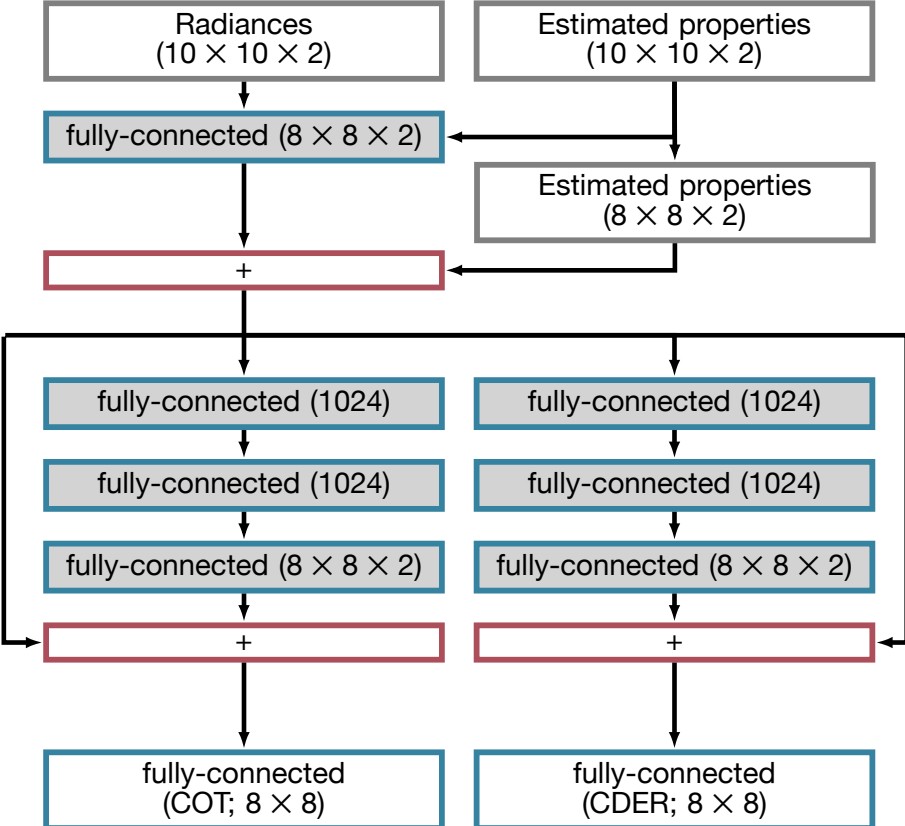

**Figure 3.** Structure of the DNN-2r network. Blue rectangles denote fully-connected layers, and a red rectangle denotes the addition of two vectors. A gray background indicates that the layer use the activation function. The numbers of units in each layer are shown in parentheses.

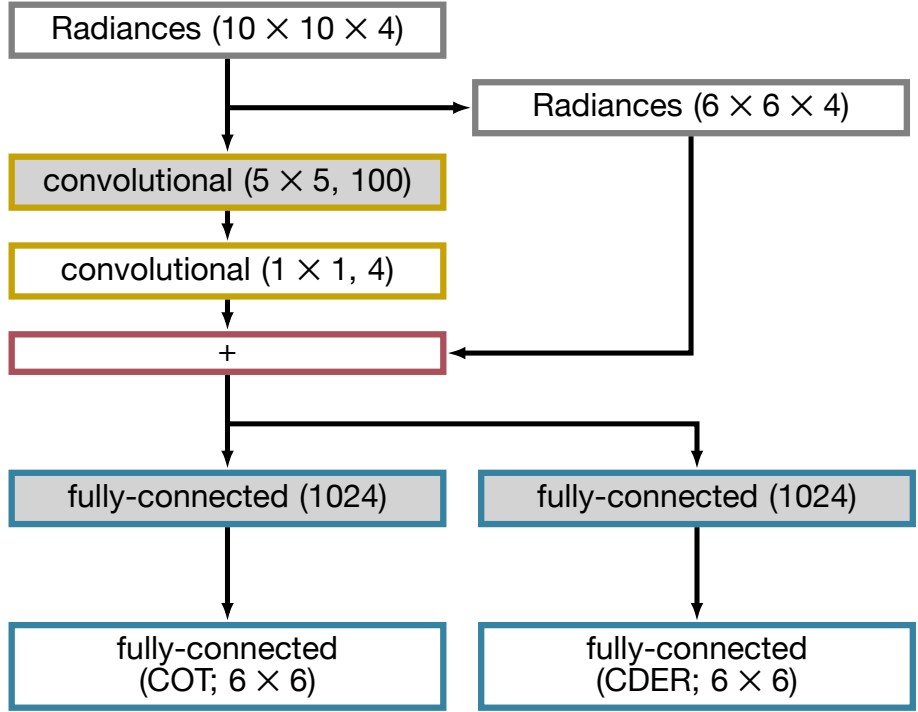

**Figure 4.** The same as Fig. 3 but for the DNN-4w network. Yellow rectangles denote the convolutional layers, for which the numbers in parentheses denote the filter size and the number of output channels. The number of filters is determined by multiplying the numbers of input and output channels.

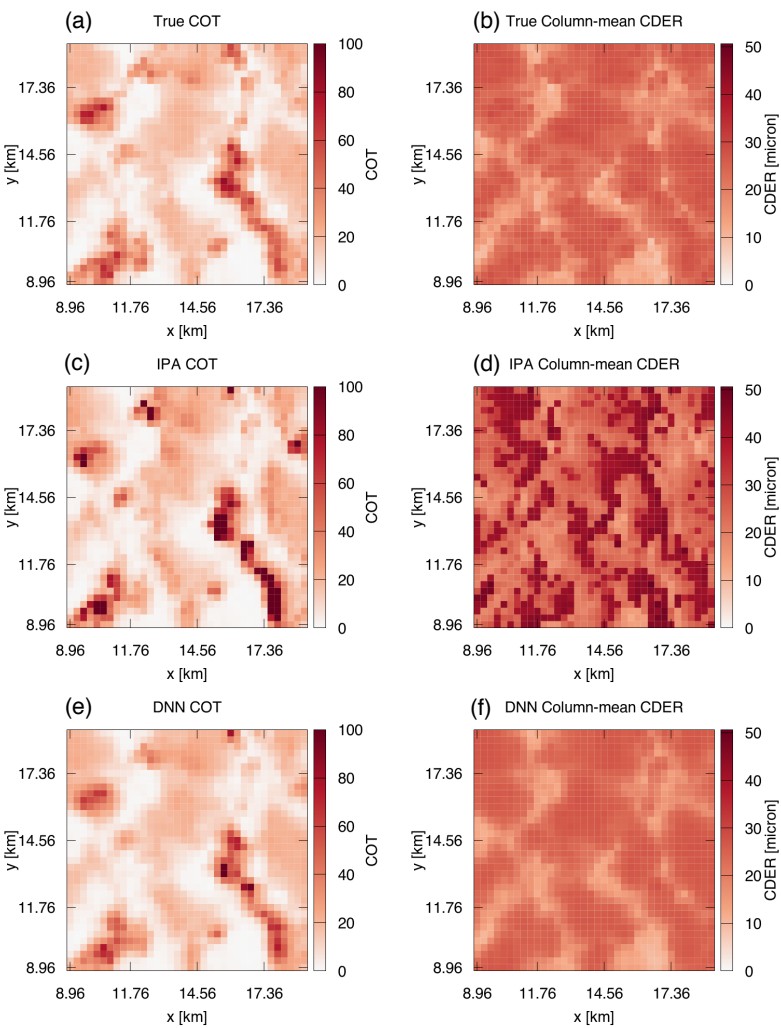

**Figure 5.** Examples of estimated (a,c,e) COT and (b,d,f) CDER of IPA and DNN retrievals for a view zenith angle of $0°$. The sun is located on the left-hand side with an SZA of $60°$. (a,b) True (reference) values of COT and CDER, (c,d) IPA retrievals, and (e,f) DNN-4w retrievals.

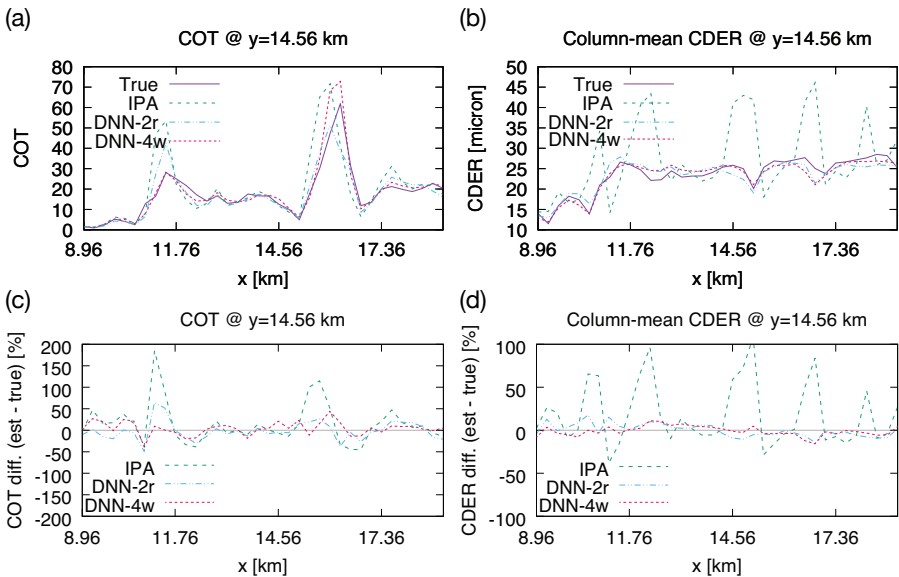

**Figure 6.** Examples of horizontal distribution of estimated (a) COT and (b) CDER by the IPA retrieval and DNNs for a view zenith angle of $0°$ at $y = 14.56$ km in Fig. 5. The relative error of the estimated (c) COT and (d) CDER. The sun is located on the left-hand side with an SZA of $60°$.

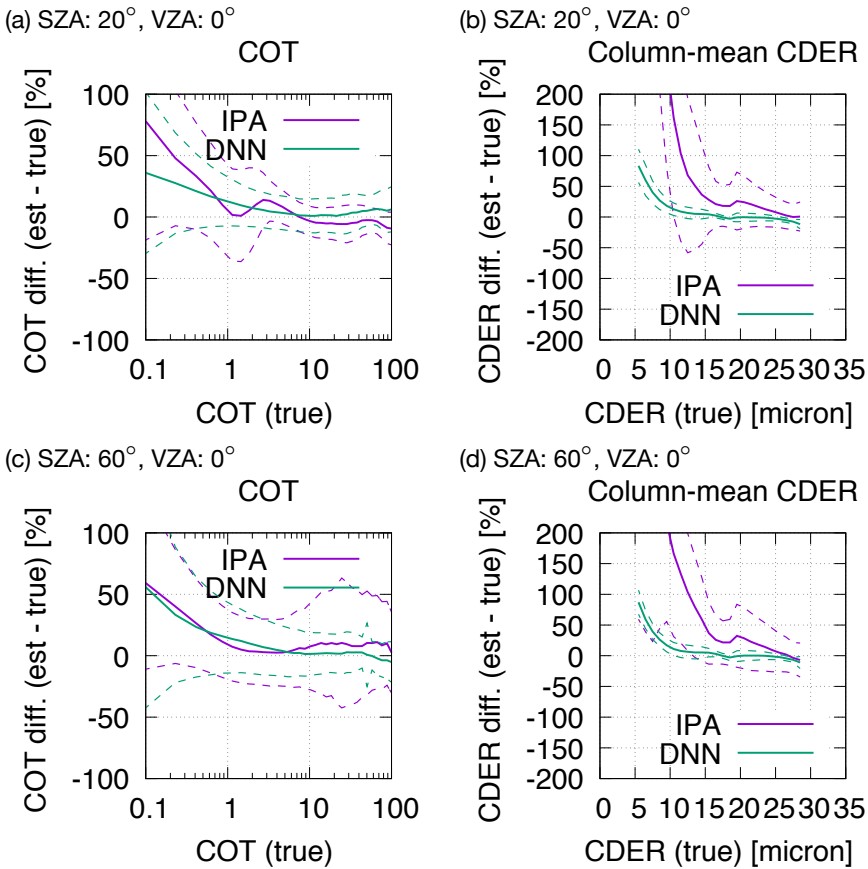

**Figure 7.** Comparison of retrieval errors of DNN-4w and IPA retrieval for SZAs of (a,b) $20°$ and (c,d) $60°$. The horizontal axes show the true values of either COT or column-mean CDER. The vertical axes show the relative error of the estimated cloud property. The solid and dashed lines denote mean errors and means plus/minus standard deviations of error, respectively.

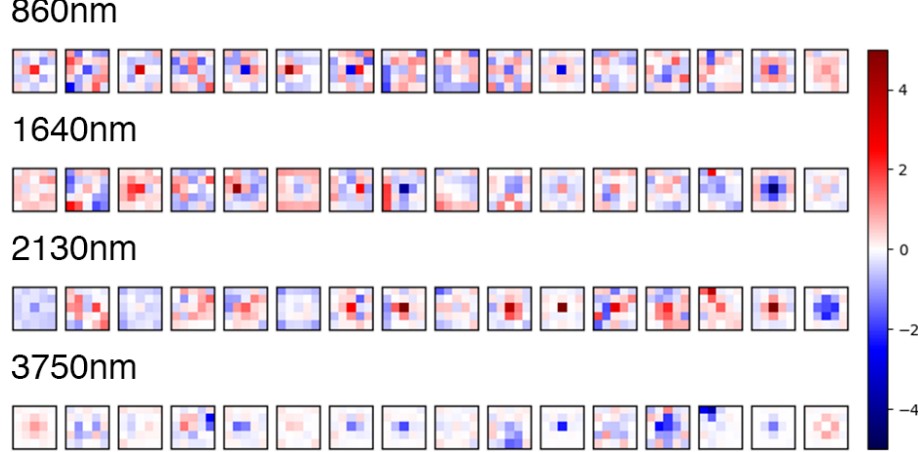

**Figure 8.** Selected examples of the filter for $5 \times 5$ pixels at each wavelength in the first convolutional layer in DNN-4w. Only 16 of 100 filters for each wavelength are shown here. The color shade denotes the filter weight. The sunlight is from the left.

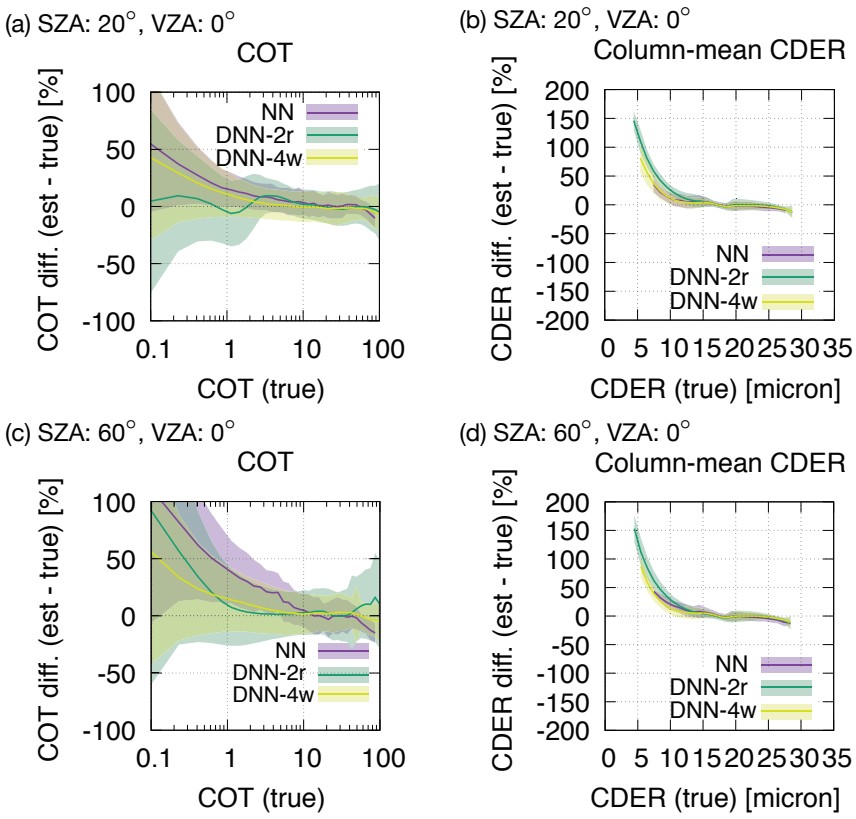

**Figure 9.** The same as Fig. 7 but for an NN and our DNNs. The solid lines show mean errors, and the shades denote regions of means plus/minus standard deviations.