# Peer review of "Feasibility study of multipixel retrieval of optical thickness and droplet effective radius of inhomogeneous clouds using deep learning"

_Atmospheric Measurement Techniques, 2017_

## Referee Comment (RC1) · Anonymous Referee #1 · 27 Jul 2017

This article presents the possibility to use of deep neural network (DNN) to retrieve cloud properties (optical thickness and effective radius) accounting for the horizontal photon transport. This is so far not accounted for in the classical algorithm, that use the homogeneous cloud assumption. This is a move in the way to improve remote sensing algorithm and account for 3D radiative effects. However, the presentation and explanations do not put the paper in favor. Consequently, several precisions and corrections need to be added in the paper before publications. They are indicated below.

Major comments:

1) The originality of the paper seems to be more related to the possibility to make a multi-pixel inversion of cloud properties than to use DNN. It should appear in the title. I suggest "Feasibility study of multi-pixels retrieval of cloud optical thickness and effective radius using deep neural network"

2) Abstract is too succinct and need to be completed.

3) In the introduction, the authors described similar works, they did previously to retrieve COT and CEDR accounting for neighboring pixels (Iwabuchi et Hayasaka, 2003). Through the paper, the disadvantages of this previous method comparing to the new one are not sufficiently explained. I did not understand "which was an obstacle to generalizing the algorithm (p2, li10)". Which obstacles? Does it not the same problem with the NN method ? The authors should add a discussion about the advantages/disadvantages and about the implementation of each method in the introduction or in the conclusion A comparison of previous method and DNN method in terms of results will also be valuable.

4) li 15-22: Some important references are incorrectly cited: Faure et al. (2001) is about the retrieval of mean cloud properties accounting for the sub-pixel heterogeneities while Faure et al. (2002) concerns the retrieval of cloud parameters from high-resolution data using adjacent pixels which is a different study. This second one is the closest to the current study. Correct it also in section 4.2. These two papers are major because they are the first papers in the fields but they are limited to fluxes and not applicable to real data. Following the paper of Faure et al., 2001, which is for medium spatial resolution, Cornet et al. (2004) present ways to apply to real data heterogeneous cloud retrieval using NN. It is finally tested on real data in Cornet et al. (2005) on MODIS data. The paragraph citing these studies about cloud neural network retrieval needs to be clarified.

5) To my knowledge, this is the first time in atmospheric science that Deep Neural Network (DNNs) are used. More explanations are needed in a specific section explaining

clearly how it works and allowing to understand some affirmations and vocabulary used in the text. For example, in the introduction, why "a DNN is more suitable for approximating complex non linear functions" than a classical NN? What is "automatic feature extraction", can the authors give an example? For the same reasons, Section 3.1 are confuse and consequently not very clear for a non-expert in deep learning. It needs to be separate with generality on the DNN in the specific section rewritten with more explanation and in a pedagogical way. Some schemas may also help to understand. Another section should specify to the choices made (see major comment 6 below). In the specific section about DNN should appears what is "shortcuts DNN"(p5, li 17) or what is convolutional layer ? How the filter weights are obtained? Can the authors also explain in few lines the paper of He et al. (2015) in order that the readers understand?

6) p5 . There is also no enough explanation about the choice of the input vector and the architecture of the DNN. Li- 5-10: why these two input vectors? The paragraph should start with an explanation of the philosophy. The first input vector is built in order to correct IPA retrieval and the second to retrieve directly cloud properties. I'm wondering also why four wavelengths and not only two as for the bi-spectral method? Does the authors test this last configuration with two wavelengths? I'm wondering also how the architecture of the DNN was chosen (convolutional layer or not, activation function or not), does the authors made test to find the best architecture?

7) P7- li 28-30: I am not completely agree with the assertion "the DNN retrieves COT values that are close to the true values assumed in the test, successfully corrected the phase lag." In Figure 6-a, near 11.7km, DNN-2r retrieval shows also large differences and near 17km clearly the DNN retrievals overestimate the COT and the phase lag is not completely cancelled. Can the authors be more precise in the description of the figures? In addition to cross-sections, could also the authors add the relative errors transects and the RMSE of the different retrievals to have more qualitative idea of the improvements. Same remarks concerning Re retrieval.

8) p8, li 1-6: Concerning Re retrieval with the homogenous cloud assumption, it is not

really surprising to obtain large differences between homogeneous assumption and true results. The overestimation with IPA is not only related with shadowing effects. Indeed, homogeneous cloud assumption involve homogeneous Re profile. From satellite remote sensing, the upper part of the cloud is retrieved (See for example Platnick et al. 2000). Therefore, if the effective radius is vertically increasing in the cloud, the retrieved Re is larger than the mean Re. For the DNN, training with heterogeneous clouds allows to learn the relation between vertically averaged Re and radiances. Discussion about this issue (p8, li 20-24) is too late in the paper and should be moved here. Try also to highlight better the shadowing by reporting for example the difference between true COT and homogeneous COT as in Cornet et al. (2015) or Marshak et al., (2006).

9) P9, section 4.2: Authors made comparisons with previous works of Faure et al., 2002 but the settings are exactly the same. First, only pairs of wavelengths were used and not the four wavelengths mentioned. In addition, in the study of Faure et al., (2002), 15 neighboring pixels of each side of the target pixels were used (62 components in the input vector) and here only 3. This can change a lot the results. The comparisons have to be done again with the same parameters than the one used in Faure et al. (2002) or at least the same conditions that the DNN, that is 10 pixels for each side, otherwise, it is not possible to conclude the comparisons and to know really why retrieval is better (DNN or neighboring pixels?)

Minor comments:

1) p. 1, li 17: why the bispectral method follows the IPA assumption? The authors should add reasons why in the text (time computation, simplicity, others?) and also insist about the independence of each cloudy columns which is considered infinite.

2) p.2, li 5: Until which distance, have the neighboring pixels to be considered ? can the authors here or further in the text give some values and references ?

3) p3, section 2.1: what are the resolution and dimension of the generated cloud fields?

4) p3, li 17: IPA (Independent Pixel Approximation) does not mean that the vertical profiles is homogeneous but only that each pixel is considered independently of his neighbors. Authors should speak about the homogeneous cloud assumption horizontally as well as vertically

5) p3, eq. 3: Why the authors used the square of the usual definition of the inhomogeneity parameter defined in the others study. For comparisons, it seems to be better to use the same definition.

6) P5, li 9: Radiances at 3.75micron is used, I suppose that is only the solar part. It should be precise in the text that thermal correction need to be done before using this wavelength.

7) p5: explain why the number of pixels considered in the input vector (10x10) is larger than those considered in the output vector (8x8 or 6x6) and why it is not the same for the two DNN. 8) P6, li10: add the URL for the chainer framework

9) P6, eq 8 and9: is there a justification for the choice of these functions?

10) P7, li 14: DNN and IPA are not really comparable: the first is an inversion tools as look-up tables and the second one is a direct model. It seems better to write multipixels-DNN inversion versus IPA-LUT inversion.

11) Figure 5 and 6: Precise data corresponds to only the test data set or to a mix between the training and test dataset

12) p7, li 22 and Figure 5 and 6: precise the geometry of the observation: view zenithal and azimuthal angles?

13) p7: li 26: illumination and shadowed effects are well-known under IPA assumption: please add some references

14) p7: li 26: Large errors are due to the flattening of the relation-ship between radiances and COT due to saturation effects: a small difference in radiance lead to a quite

large difference in COT.

15) p8, li 9, figure 7 : I agree that the bias (mean error) is particularly large only for COT less than 1. For COT > 1, the difference in errors is not so important. For COT >10, the standard deviation is larger for IPA meaning that dispersion (roughening) is more important.

16) Figure 8: Could the authors indicate the COT and Re associated with the filters and be more precise in the description of the figure? Which filters patterns are "symmetrical around the center" and how is distributed the optical thickness? Also on which figures does appear "the feature related to the solar direct beam"? Comments also the difference between wavelengths.

17) Section 4.2: are the same training set and generalization set used for all the NN trainings?

18) p9, li 30-33: add the issue concerning the vertical profiles for Re in the conclusion.

19) P10: in the conclusion, can the authors insist on the limitations of using NN methods such as the one related to database used and extrapolation issues. In other words, how will work the DNN is the cloud is quite different to those used for the training dataset?

20) P10: Following the previous points, can the authors speaks about the steps needed in order to develop an operational multi-pixels algorithm?

References: Cornet, C., Buriez, J-C., Riédi, J., Isaka, H. and B. Guillemet, 2005: Case study of inhomogeneous cloud parameter retrieval from MODIS data, Geophys. Res. Lett, 32, L13807, doi:10.1029/2005GL022791.

Alexander Marshak, Steven Platnick, Tamás Várnai, Guoyong Wen, Robert F. Cahalan, Impact of three-dimensional radiative effects on satellite retrievals of cloud droplet sizes, Journal of Geophysical Research, 2006, 111, D9

---

## Referee Comment (RC2) · Z. Zhang (Referee) · 3 Aug 2017

Comments on "Retrieval of optical thickness and droplet effective radius of inhomogeneous clouds using deep learning" by Okamura et al.

This paper documents a retrieval algorithm based on the deep learning neural network (DNN) for retrieving the cloud optical thickness and cloud effective radius from spectral cloud reflectance observations. The DNN algorithm is trained by synthetic cloud fields from LES and simulated cloud reflectances using 3D radiative transfer models. It is shown that a great advantage of the DNN algorithm is its apparent immunity to the so-called 3-D radiative transfer effects. The "traditional" Look-up-table method suffers from

significant biases due to the illuminating and shallowing effects, while the retrievals from the DNN algorithm are less affected by these biases and agree better with the "ground truth" from LES.

Overall, I found this paper interesting and exciting, and certainly suitable for AMT. On the other hand, I do have a few questions/suggestions that are listed below and hope they can help the author further improve the paper. For full disclosure, I know almost nothing about neural network or machine learning. So, my comments will be mostly from the perspective of cloud remote sensing which is my research field.

1) Robustness of the results: I'm very excited to see that the DNN-based algorithm is able to overcome the influence of 3-D effects (illuminating and shadowing) and yield retrievals in close agreement on LES. My biggest concern is that if this result is robust enough. I hope I am not mistaken, but it seems the DNN in this study is trained using only two LES cases in Figure 1 and moreover only applied to these two cases. If so, frankly, I not completely convinced if the algorithm will generate same successful retrievals if it is applied to other LES scenes or real satellite images. To convince me and the readers, the authors should consider a "blind test", in which they apply the algorithm to the LES cases other than the two training cases. For example, the authors can tweak the meteorological conditions in the LES (e.g., inversion strength, large-scale forcing etc.) to generate different cloud types/scenes, and then apply the DNN algorithm to assess and report its performance. r Overall, the authors need to demonstrate the robustness of their algorithm and results.

2) Complexity of the training: Note that 3-D radiative effects depend on many factors, not only just COT, CER and solar geometry, but also cloud top inhomogeneity, cloud geometrical thickness and surface reflectance among others [Várnai and Davies, 1999; Várnai and Marshak, 2001; 2002] as well as instrument characteristics. I'm wondering which ones of these factors have to be part of the training and which ones do not need to be. Take surface reflectance for example. Can we train the algorithm using only one surface reflectance and then it will work for all other types of surface? In addition to 3-D

effects, the retrievals are also affected by many other factors, the presence of drizzle, atmospheric absorption, surface reflectance etc. It is not clear from the paper to what extent these factors are considered in the DNN algorithm training, and which ones are not. Overall, I'm trying to figure out how "smart" the algorithm is. If we have to worry about all the above-mentioned details in the training, then the practical usefulness of the algorithm becomes questionable.

3) Cloud mask: It is not clear from the paper how cloud masking is treated in the retrieval/training. If retrievals are done at the resolution coarser than the LES grid, then some pixels are inevitably partly cloudy. How are the partly cloudy pixels treated in the retrievals and training?

4) Definitions of CER: When cloud microphysics varies both vertically and horizontally, then the definition of CER can be very tricky. For example, Eq. (1) applies well to a single LES cell, no problem. (the root and meaning of the parameter need to be explained in detail though). The equation (2) for column-mean CER becomes tricky. First of all, does the vertical average takes into account any vertical weighting for example due to photon penetration depth [Platnick, 2000; Miller et al., 2016]? Some explanations are needed either way. Second, what the column-mean ? How to compute it? Third, what is the significance of the column-mean CER in Eq. (2)? Does it help understand the cloud radiative effects? Does it help the modelers validate their cloud microphysics simulations? Can it be used in combination of COT retrieval to estimate LWP?

After defining the column-mean CER for a single column, the authors also need to explain how to aggregate/define the CER over multiple LES columns horizontally. For example, if the retrievals are done at 10x10 pixels, and each pixel has a slightly different column-mean CER, then what is the CER for the 10x10 pixel ensemble?

There are a few recent studies that discussed this topic. Maybe they are helpful [Miller et al., 2016] and [Alexandrov et al., 2012]

5) Plane-parallel albedo bias: This study focuses on the impacts caused by IPA, but

there is another type of bias, plane-parallel-albedo bias (PPHB). It is not clear to me if the DNN described in this study could also take care of the PPHB. Note that recently, Zhang et al. [Zhang et al., 2016] described a novel method to correct the PPHB, which might be helpful for this study.

6) Lack of technique details: I agree with the other reviewer that many important technique details are lacking from the current paper. Currently, the paper is rather short, so there is plenty of space to add in more detailed description and discussion, especially for Section 3 Method. Just to give an example, what are the meaning of Eq. (8) and (9)? Why do they provide the "relationships between inputs and outputs variables" of DDN, what kind of relationship?

Alexandrov, M. D., B. Cairns, C. Emde, A. S. Ackerman, and B. van Diedenhoven (2012), Accuracy assessments of cloud droplet size retrievals from polarized reflectance measurements by the research scanning polarimeter, Remote Sensing of Environment, 125(0), 92–111, doi:10.1016/j.rse.2012.07.012.

Miller, D. J., Z. Zhang, A. S. Ackerman, S. Platnick, and B. A. Baum (2016), The impact of cloud vertical profile on liquid water path retrieval based on the bispectral method: A theoretical study based on large‐eddy simulations of shallow marine boundary layer clouds, Journal of Geophysical Research-Atmospheres, 121(8), 4122–4141, doi:10.1002/2015JD024322.

Platnick, S. (2000), Vertical photon transport in cloud remote sensing problems, Journal of Geophysical Research, 105(D18).

Várnai, T., and A. Marshak (2001), Statistical Analysis of the Uncertainties in Cloud Optical Depth Retrievals Caused by Three-Dimensional Radiative Effects, http://dx.doi.org/10.1175/1520-0469(2001)058<1540:SAOTUI>2.0.CO;2, 58(12), 1540–1548, doi:10.1175/1520-0469(2001)058<1540:SAOTUI>2.0.CO;2.

Várnai, T., and A. Marshak (2002), Observations of Three-Dimensional Radiative Effects that Influence MODIS Cloud Optical Thickness Retrievals, J. Atmos. Sci., 59(9), 1607–1618, doi:10.1175/1520-0469(2002)059<1607:OOTDRE>2.0.CO;2.

Várnai, T., and R. Davies (1999), Effects of Cloud Heterogeneities on Shortwave Radiation: Comparison of Cloud-Top Variability and Internal Heterogeneity, J. Atmospheric Sciences, 56(24), 4206–4224, doi:10.1175/1520-0469(1999)056<4206:EOCHOS>2.0.CO;2.

Zhang, Z., F. Werner, H. M. Cho, G. Wind, S. Platnick, A. S. Ackerman, L. Di Girolamo, A. Marshak, and K. Meyer (2016), A framework based on 2-D Taylor expansion for quantifying the impacts of sub-pixel reflectance variance and covariance on cloud optical thickness and effective radius retrievals based on the bi-spectral method, Journal of Geophysical Research-Atmospheres, 2016JD024837, doi:10.1002/2016JD024837.

---

## Referee Comment (RC3) · Anonymous Referee #2 · 25 Aug 2017

The paper describes a new technique for satellite measurements of cloud optical thickness and cloud droplet effective radius. The key feature of the technique is that it takes into account 3D radiative effects and subpixel variability by considering not one pixel at a time, but by performing simultaneous retrievals over 10 by 10 pixel areas. The most important aspect of the technique is the use of a deep learning algorithm. This is a significant new development, and the study makes an important contribution on the path toward more accurate satellite retrievals of cloud properties. Overall, the methodology is sound and the presentation is suitable. However, I believe that a few important improvements are needed in the analysis. My recommendation is therefore to make some major revisions. Please find below my detailed comments.

[Figure]

Major issues:

1.

Page 7, Line 8 mentions that "The test dataset used for evaluation should be independent of the training dataset." My sense is that in this initial study the training and testing datasets are not fully independent, as they come from the very same cloud fields, and that this would be good to mention. (The two datasets include different randomly selected locations within the cloud fields, but the statistics of cloud properties are identical in the training and testing datasets.)

As noted in Page 10, Lines 7-8, it will be an important future step to examine the performance of the retrieval for a wider range of cloud parameters. It is reasonable to leave this (and the evaluation based on fully independent training and testing datasets) to a future paper, but even the current results could offer further insights into the robustness of the proposed retrieval algorithms. Most importantly, one could examine not only the overall results, but also separately the results for open-cell and closed-cell convection cases. This would demonstrate that the same algorithm and training set improves retrieval accuracy for two very different types of cloud structures. I don't think the currently presented results show this: Overall error statistics may conceivably improve due to improvements for open-cell convection only, without any improvements for closed-cell convection. (Because retrieval uncertainties are likely larger for open-cell convection, it may be best to examine by what percentage DNN-2r and DNN-4w reduce the retrieval errors of IPA retrievals for open-cell and for closed-cell convection.) The paper did a good job in examining results as a function of optical thickness, but the new analysis of already performed retrievals would help because open-cell and closed-cell convection cases differ in horizontal structure even at locations where vertical optical thicknesses are similar.

2.

Page 5, Lines 13 and 22: I wonder why scene parameters are estimated for 8 X 8

pixel arrays when using the DNN-2r method, but only for the central 6 X 6 pixel arrays when using the DNN-4w method. This could make sense if 3D effects acted over larger distances at 3.75 microns than at the 0.86 and 2.15 microns used by the DNN-2r method, but neither my own physical reasoning nor the filter weights in Figure 8 suggest this. In fact, Figure 8 shows that DNN-4w retrievals at a pixel are strongly affected by 0.86 micron radiances 2 pixels away. This suggests that (at least for pixels at the edges of 8 X 8 pixel areas) the DNN-2r method cannot capture the portion of 3D effects caused by areas more than a pixel away. This probably contributes to DNN-2r giving less accurate results than DNN-4w (a tendency mentioned in Page 9, Lines 31-32) and should be mentioned in the discussion of the differences between the two methods at the top of Page 10. (The discussion should also include the impact of additional wavelengths in DNN-4w and algorithmic differences.) Also, it could help to clarify explicitly in the paper whether DNN-2r retrievals inside (not along the edges of) 8 X 8 pixel areas are affected by radiances 2 pixels or more away. If they were, it could even make sense to analyze retrieval accuracy only for pixels in the central 6 X 6 pixels of 10 X 10 pixel areas (similarly to DNN-4w).

Minor issues:

Page 1, Line 23: What is meant by "cloud state"?

Page 2, Line 23: The study by Evans et al. ("The Potential for Improved Boundary Layer Cloud Optical Depth Retrievals from the Multiple Directions of MISR", J. Atmos. Sci., 2008) should also be mentioned, as it also used a neural net for cloud retrievals.

Page 2, Lines 26-27: What is meant by "feature" and "feature extraction"?

Figure 2: It would help to indicate the time elapsed during the 60 time steps along the horizontal axis, or to mention the time step in the figure caption.

Figure 3: It would help to clarify why there is a fully connected layer near the top of the left column that operates only on radiances and not on the IPA-estimated scene

parameters.

Page 6, Lines 27-29: It would help to clarify whether all pixels within an LES scene are multiplied by the same randomly chosen number, or all individual pixels are multiplied by a different number. (My guess is the first option.)

Page 6, Line 25 and Page 7, Lines 8-10: What does the word "samples" refer to? My guess is that each sample is a 10 X 10 pixel area. If my guess is right, can samples overlap? Also, it would help to mention the total number of pixels in the LES dataset, as this could show whether the training set includes almost all LES pixels or just a small fraction of them.

Figure 7: It could help to include into one of the panels a PDF of true optical thickness values.

Figure 9: The legend should indicate which color shading corresponds to which line/method.

Page 10, Lines 5-6: I am not sure the sentence "In the DNN-4w that we tested, we excluded 3D radiative transfer effects that occurred at horizontal scales greater than approximately 1.5 km (5 pixels)" is correct. Based on Figure 8, I thought that DNN-4w retrievals exclude 3D effects that occur at horizontal scales greater than 2 pixels (560 m). This is because I thought the pixel whose properties we are retrieving is at the center of the filters in Figure 8, which means that only radiances two pixels away are considered. A correction of this sentence or a clarification of the meaning of filters in Figure 8 would help.

Somewhere in the text it would help to comment on whether the speed of calculations would be a concern for using DNN in operational retrievals in the near future. (For example, how does the speed of DNN compare to the speed of IPA and NN retrievals?)

---

## Author Comment (AC3) · 31 Oct 2017

Please find the supplement for our responses.

Please also note the supplement to this comment:
https://www.atmos-meas-tech-discuss.net/amt-2017-154/amt-2017-154-AC3-supplement.pdf

———————————————————

---

## Author Response (AR1)

**Responses to reviews**

**Reply to anonymous referee 1**

We thank the reviewer for the time and efforts she/he spent reading our manuscript carefully and providing valuable comments and suggestions. In the revised manuscript, we have tried to accommodate all the suggested changes. Please find below our responses. The reviewer's comments are in italic. Changes/additions made to the text are given in quotes.

*This article presents the possibility to use of deep neural network (DNN) to retrieve cloud properties (optical thickness and effective radius) accounting for the horizontal photon transport. This is so far not accounted for in the classical algorithm, that use the homogeneous cloud assumption. This is a move in the way to improve remote sensing algorithm and account for 3D radiative effects. However, the presentation and explanations do not put the paper in favor. Consequently, several precisions and corrections need to be added in the paper before publications. They are indicated below.*

*Major comments*

*1) The originality of the paper seems to be more related to the possibility to make a multi-pixel inversion of cloud properties than to use DNN. It should appear in the title. I suggest "Feasibility study of multi-pixels retrieval of cloud optical thickness and effective radius using deep neural network"*

**Response**: We appreciate the suggestion. Indeed, this is a feasibility study of such a multi-pixel inversion. It is also important that the deep learning techniques are applied to retrieval of inhomogeneous clouds. Thus, by combining the reviewer's suggestion and our points, we have changed the title as
"Feasibility study of multipixel retrieval of optical thickness and droplet effective radius of inhomogeneous clouds using deep learning"

*2) Abstract is too succinct and need to be completed.*

**Response**: We have rewritten the abstract, adding explanations about the two DNNs

*3) In the introduction, the authors described similar works, they did previously to retrieve COT and CEDR accounting for neighboring pixels (Iwabuchi et Hayasaka, 2003). Through the paper, the disadvantages of this previous method comparing to the new*

*one are not sufficiently explained. I did not understand "which was an obstacle to generalizing the algorithm (p2, li10)". Which obstacles? Does it not the same problem with the NN method ? The authors should add a discussion about the advantages/disadvantages and about the implementation of each method in the introduction*
*or in the conclusion A comparison of previous method and DNN method in terms of results will also be valuable.*

**Response**: We have rewritten explanations about limitations in the previous study by Iwabuchi and Hayasaka (2003), as follows:

"Since 3D radiative effects vary with COT, CDER, cloud geometrical thickness, cloud top roughness, and sun–cloud–satellite geometry, Iwabuchi and Hayasaka (2003) had to construct different sets of fitting coefficients, thus limiting the applicability of the technique in practice. In addition, their method was based on linear regression, which is not flexible to account for any nonlinear 3D radiative transfer effects."

A comparison with the previous method in Iwabuchi et Hayasaka (2003) is technically difficult. One reason is that the previous method was developed for a feasibility study assuming a particular type of cloud model (Fourier transform cloud), which is quite different from that in the present study. In addition, the assumed cloud inhomogeneity was in one horizontal dimension while the present study treats two horizontal dimensions.

We have added an explanation of one of advantage of DNN, ease of addition of input and output variables, as follows:
" In addition, input and output parameters can easily be added, and structures can be modified - another advantage of deep learning"

*4) li 15-22: Some important references are incorrectly cited: Faure et al. (2001) is about the retrieval of mean cloud properties accounting for the sub-pixel heterogeneities while Faure et al. (2002) concerns the retrieval of cloud parameters from high-resolution data using adjacent pixels which is a different study. This second one is the closest to the current study. Correct it also in section 4.2. These two papers are major because they are the first papers in the fields but they are limited to fluxes and not applicable to real data. Following the paper of Faure et al., 2001, which is for medium spatial resolution, Cornet et al. (2004) present ways to apply to real data heterogeneous cloud retrieval using NN. It is finally tested on real data in Cornet et al. (2005) on MODIS data. The paragraph citing these studies about cloud neural network retrieval needs to be clarified.*

**Response**: As the reviewer requested, we revised Section 1 and clarified the description about

previous studies. The main discussion in Faure et al. (2001) was the application of NN to retrieval of mean cloud properties, taking sub-pixel inhomogeneity into account, but they also tested a NN that uses the radiances of target pixel (0.8km x 0.8km region) and eight adjacent pixels (0.8km x 0.8km regions) for retrievals, showing a promising result. Our DNN method is compared to the latter one in the Faure et al.'s (2001) study. We have added descriptions about this point in the Section 4.2, as follows:

" Originally, this NN had two hidden layers with 10 units each and it used the 0.8 km $\times$ 0.8 km area-averaged radiances at four wavelengths (0.64, 1.6, 2.2, 3.7 $\mu$m) at the target pixel and eight adjacent pixels (called as "with ancillary data"). It is described in the section 3.3 (2) in the original paper."

Faure et al. (2002) uses high resolution but one-dimensional cloud data, and we cannot directly compare our results with that of Faure et al. (2002). We have added the reference Cornet et al. (2005) in Section 1.

*5) To my knowledge, this is the first time in atmospheric science that Deep Neural Network (DNNs) are used. More explanations are needed in a specific section explaining clearly how it works and allowing to understand some affirmations and vocabulary used in the text. For example, in the introduction, why "a DNN is more suitable for approximating complex non linear functions" than a classical NN? What is "automatic feature extraction", can the authors give an example? For the same reasons, Section 3.1 are confuse and consequently not very clear for a non-expert in deep learning. It needs to be separate with generality on the DNN in the specific section rewritten with more explanation and in a pedagogical way. Some schemas may also help to understand. Another section should specify to the choices made (see major comment 6 below). In the specific section about DNN should appears what is "shortcuts DNN"(p5, li 17) or what is convolutional layer ? How the filter weights are obtained? Can the authors also explain in few lines the paper of He et al. (2015) in order that the readers understand?*

**Response**: We appreciate this comment. We have moved explanations on fundamental DNN techniques to Section 3.1. As the reviewer commented, some important explanations were missing in the previous manuscript. So we have added some more explanations (ex. "a DNN is more suitable for approximating complex problem", "automatic feature extraction", and "shortcuts"). We hope the revised manuscript is easier to read.

We have not added very long explanations on the techniques that are really technical and not essential to the conclusions of this paper. We did discuss essential characteristics of each deep learning technique in the current manuscript. We think it is better to leave the technical details of each optimization and deep learning technique for readers to consult the textbooks or references cited in the current manuscript. Indeed, the deep learning is applied to a remote

sensing problem of atmospheric target for the first time, to our best knowledge. However, it is a rapidly growing technique in broad areas of sciences, upon many successes in engineering and applications for the artificial intelligence. In the atmospheric sciences, too, we know at least a few research groups working on applications of deep learning techniques. Nowadays, it is easy to find a book for practical use of this technology even for undergraduate students.

*6) p5 . There is also no enough explanation about the choice of the input vector and the architecture of the DNN. Li- 5-10: why these two input vectors? The paragraph should start with an explanation of the philosophy. The first input vector is built in order to correct IPA retrieval and the second to retrieve directly cloud properties. I'm wondering also why four wavelengths and not only two as for the bi-spectral method? Does the authors test this last configuration with two wavelengths? I'm wondering also how the architecture of the DNN was chosen (convolutional layer or not, activation function or not), does the authors made test to find the best architecture?*

**Response**: As the reviewer commented, we tested numbers of different DNNs with different number of wavelengths, the use of convolutional layer, different activation function, and so on. The best two DNNs are shown in this paper. We have added the description about this in the manuscript, as follows:
"The above two DNN structures were obtained from various trial-and-error experiments. Different DNN structures were also tested. For example, we tested a DNN similar to DNN-2r but with four wavelengths, and one similar to DNN-4w but with only two wavelengths. However, DNN-2r and DNN-4w performed best. There is room for improvement in DNN structures, which should be investigated in the future."

*7) P7- li 28-30: I am not completely agree with the assertion "the DNN retrieves COT values that are close to the true values assumed in the test, successfully corrected the phase lag." In Figure 6-a, near 11.7km, DNN-2r retrieval shows also large differences and near 17km clearly the DNN retrievals overestimate the COT and the phase lag is not completely cancelled. Can the authors be more precise in the description of the figures? In addition to cross-sections, could also the authors add the relative errors transects and the RMSE of the different retrievals to have more qualitative idea of the improvements. Same remarks concerning Re retrieval.*

**Response**: We have rewritten the sentence as
"Compared to the IPA retrieval, the DNNs retrieve COT values that are closer to the true values assumed in this test. The DNN-4w successfully corrects the phase lag as shown in Fig. 6c."
As the reviewer suggested, we have added figures of cross-sections of the relative errors and a table of RMSEs of the different retrievals in Figure 6.

*8) p8, li 1-6: Concerning Re retrieval with the homogenous cloud assumption, it is not really surprising to obtain large differences between homogeneous assumption and true results. The overestimation with IPA is not only related with shadowing effects. Indeed, homogeneous cloud assumption involve homogeneous Re profile. From satellite remote sensing, the upper part of the cloud is retrieved (See for example Platnick et al. 2000). Therefore, if the effective radius is vertically increasing in the cloud, the retrieved Re is larger than the mean Re. For the DNN, training with heterogeneous clouds allows to learn the relation between vertically averaged Re and radiances. Discussion about this issue (p8, li 20-24) is too late in the paper and should be moved here. Try also to highlight better the shadowing by reporting for example the difference between true COT and homogeneous COT as in Cornet et al. (2015) or Marshak et al., (2006).*

**Response**: Figures 5 and 6 are good for demonstrating spatial characteristics of retrieval errors, but the mean bias error for all test data are shown in Fig. 7. Differences between vertically homogeneous and inhomogeneous assumptions are well related to the mean bias in Re. We think it is better to see the discussion about the issue as in the current manuscript. We have modified the sentences as follows:

"IPA-retrieved CDER thus tends to be larger than column-mean CDER, whereas DNNs are by design trained to learn the relationship between the column-mean CDER and radiances by taking into account the vertical inhomogeneity. However, this vertical inhomogeneity effect on the IPA-retrieved CDER does not seem to be the main cause of the large positive bias."

The issue of difference between true COT and homogeneous COT as in Cornet et al. (2015) should be important, but the sub-pixel horizontal inhomogeneity is not considered in this paper.

*9) P9, section 4.2: Authors made comparisons with previous works of Faure et al., 2002 but the settings are exactly the same. First, only pairs of wavelengths were used and not the four wavelengths mentioned. In addition, in the study of Faure et al., (2002), 15 neighboring pixels of each side of the target pixels were used (62 components in the input vector) and here only 3. This can change a lot the results. The comparisons have to be done again with the same parameters than the one used in Faure et al. (2002) or at least the same conditions that the DNN, that is 10 pixels for each side, otherwise, it is not possible to conclude the comparisons and to know really why retrieval is better (DNN or neighboring pixels?)*

**Response**: As explained in the reply to the comment 4, we compared our results with that of Faure et al. (2001).

*1) p. 1, li 17: why the bispectral method follows the IPA assumption? The authors should add reasons why in the text (time computation, simplicity, others?) and also insist about the independence of each cloudy columns which is considered infinite.*

**Response**: We have rewritten the sentences as follows:

" The method is based on the independent pixel approximation (IPA) assuming plane- parallel, horizontally and vertically homogeneous cloud for each pixel of the satellite image because of high computational cost for simulation of three-dimensional (3D) radiative transfer. The observed cloud radiances result from three-dimensional (3D) radiative transfer in the cloud field, …"

*2) p.2, li 5: Until which distance, have the neighboring pixels to be considered ? can the authors here or further in the text give some values and references ?*

**Response**: We have added an explanation about the scales, as follows:

"The 3D radiative effects operate on horizontal scales that are determined mainly by cloud thickness and solar zenith angle. When the sun is oblique (i.e., with a solar zenith angle of 60◦ or larger), the maximum horizontal scale for 3D radiative effects is roughly 15–20 times larger than cloud thickness (Marshak and Davis, 2005).

*3) p3, section 2.1: what are the resolution and dimension of the generated cloud fields?*

**Response**: We added a description about the resolution of the generated cloud fields as "The resolution for the x- and y-axis of the cloud field is originally 35 m. For the z-axis, the resolution is 5 m at the bottom of the atmosphere, and it is coarse (less than 60 m) for the upper layers."
The dimension sizes are written in the manuscript.

*4) p3, li 17: IPA (Independent Pixel Approximation) does not mean that the vertical profiles is homogeneous but only that each pixel is considered independently of his neighbors. Authors should speak about the homogeneous cloud assumption horizontally as well as vertically*

**Response**: The IPA retrieval usually assumes vertically and horizontally homogeneous cloud in each pixel. We modified the introduction part as "the independent pixel approximation (IPA) assuming plane-parallel, horizontally and vertically homogeneous cloud for each pixel of the

satellite image".

*5) p3, eq. 3: Why the authors used the square of the usual definition of the inhomogeneity parameter defined in the others study. For comparisons, it seems to be better to use the same definition.*

**Response**: This is because the 3D radiative effects (e.g. 3D minus IPA radiances) are approximately linear to the square of the usual definition of the inhomogeneity. In this paper, we tried to show that the open cell cloud was much more inhomogeneous.

*6) P5, li 9: Radiances at 3.75micron is used, I suppose that is only the solar part. It should be precise in the text that thermal correction need to be done before using this wavelength.*

**Response**: We have added an explanation about this, as follows:
"It should be noted that only solar radiation is considered in the present study, which requires that the thermal radiation at 3.75 $\mu$m be corrected during pre-processing."

*7) p5: explain why the number of pixels considered in the input vector (10x10) is larger than those considered in the output vector (8x8 or 6x6) and why it is not the same for the two DNN. 8) P6, li10: add the URL for the chainer framework*

**Response**: We have added a sentence for the reason as follows:
"The reason for including margin pixels in the input field is to take into account the 3D radiative effects from the surroundings of the cloud field. The DNN-2r network consists of several fully connected layers."

*9) P6, eq 8 and9: is there a justification for the choice of these functions?*

**Response**: The functions in these equations are less nonlinear to radiances. This kind of simple transform helps better performance of DNN retrievals. We have added an introductory sentence as follows:
"Because the radiances are highly nonlinear with respect to the COT and CDER, it is convenient to transform the COT and CDER by some simple functions. "

*10) P7, li 14: DNN and IPA are not really comparable: the first is an inversion tools as look-up tables and the second one is a direct model. It seems better to write multipixelsDNN inversion versus IPA-LUT inversion.*

**Response**: We agree, but since we described our terms, we prefer to stick with the terms of "DNN and IPA retrievals". We modified the manuscript to use "IPA retrieval" in a consistent way.

*11) Figure 5 and 6: Precise data corresponds to only the test data set or to a mix between the training and test dataset*

**Response**: These are for test data.

*12) p7, li 22 and Figure 5 and 6: precise the geometry of the observation: view zenithal and azimuthal angles?*

**Response**: We added the geometry information to Figure 5 and 6, and in the manuscript.

*13) p7: li 26: illumination and shadowed effects are well-known under IPA assumption: please add some references*

**Response**: Várnai and Davies (1999), Várnai and Marshak (2002), and Marshak et al. (2006) are added as references about illuminating and shadowing effects.

*14) p7: li 26: Large errors are due to the flattening of the relation-ship between radiances and COT due to saturation effects: a small difference in radiance lead to a quite large difference in COT*

**Response**: We appreciate this comment and have added a sentence as
"This can be ascribed to weaker sensitivity of radiance to COT when COT is larger; a small difference in radiance leads to large difference in retrieved COT."

*15) p8, li 9, figure 7 : I agree that the bias (mean error) is particularly large only for COT less than 1. For COT > 1, the difference in errors is not so important. For COT>10, the standard deviation is larger for IPA meaning that dispersion (roughening) is more important.*

**Response**: Large IPA error dispersion for COT > 10 are shown for SZA of 60 degrees, which suggests that the roughening cause the error. This point is mentioned in the manuscript.

*16) Figure 8: Could the authors indicate the COT and Re associated with the filters and be more precise in the description of the figure? Which filters patterns are "symmetrical around the center" and how is distributed the optical thickness? Also on which*

*figures does appear "the feature related to the solar direct beam"? Comments also the difference between wavelengths.*

**Response**: In general, this kind of analysis of NN coefficients is difficult to understand when the NN becomes deeper. To study the associations of these filters with COT and CDER, we should investigate the coefficients in the DNN-4w layers subsequent to the convolutional layers (as shown in Fig. 4). It would be hard to interpret the relationships between the two destination fully connected layers in the DNN-4w. In this paper, we just tried to check that the DNN-4w indeed learned meaningful patterns of 3D radiative transfer.

As suggested by the reviewer, we have added a few explanations about specific pattern shown in Fig. 8. We have also added an introductory note as follows:
"It is noted that the convolutional layer is designed to correct the 3D radiative effects appeared in radiances (Fig. 4)."

*17) Section 4.2: are the same training set and generalization set used for all the NN trainings?*

**Response**: Yes, the original dataset are the same as those used for DNNs. We have added a sentence as "Data used for training and test for the NN are from the same original datasets used for DNNs."

*18) p9, li 30-33: add the issue concerning the vertical profiles for Re in the conclusion.*

**Response**: As shown in Section 4.1, the vertical profile for Re seems to be not a main reason for the IPA retrieval bias in Re, at least, in the present cases. We think there is no strong need to mention this point in the conclusion.

*19) P10: in the conclusion, can the authors insist on the limitations of using NN methods such as the one related to database used and extrapolation issues. In other words, how will work the DNN is the cloud is quite different to those used for the training dataset?*

**Response**: It is a well-known issue of NN as the reviewer suggested. Training of NN should includes enough variety of realistic cloud. This study is just a feasibility study, but practical applications definitely require training for various types of cloud. We have added a few words as "...will require training using realistic cloud fields for various types of cloud."

*20) P10: Following the previous points, can the authors speaks about the steps needed*

*in order to develop an operational multi-pixels algorithm?*

**Response**: An expansion of cloud variety is one step. An appropriate DNN architecture for addition of input parameters should be investigated. For example, the sun-cloud-satellite geometry parameters are very different from the radiance image data that are used as input data in this study. There should be appropriate DNN structure to add the geometrical parameters. We have added the last sentence in the conclusion, as "An appropriate DNN architecture for addition of input parameters should be investigated in the future."

Reply to Dr. Z. Zhang

First of all, we would like to thank the reviewer for reading our paper carefully and providing constructive comments. In the revised manuscript, we have tried to accommodate all the suggested changes. The modifications from the originally submitted version are highlighted in the revised manuscript. Please see our specific responses below.

*Comments on "Retrieval of optical thickness and droplet effective radius of inhomogeneous clouds using deep learning" by Okamura et al.*
*This paper documents a retrieval algorithm based on the deep learning neural network (DNN) for retrieving the cloud optical thickness and cloud effective radius from spectral cloud reflectance observations. The DNN algorithm is trained by synthetic cloud fields from LES and simulated cloud reflectances using 3D radiative transfer models. It is shown that a great advantage of the DNN algorithm is its apparent immunity to the so-called 3-D radiative transfer effects. The "traditional" Look-up-table method suffers from significant biases due to the illuminating and shallowing effects, while the retrievals from the DNN algorithm are less affected by these biases and agree better with the "ground truth" from LES.*

*Overall, I found this paper interesting and exciting, and certainly suitable for AMT. On the other hand, I do have a few questions/suggestions that are listed below and hope they can help the author further improve the paper. For full disclosure, I know almost nothing about neural network or machine learning. So, my comments will be mostly from the perspective of cloud remote sensing which is my research field.*

*comments*

*1) Robustness of the results: I'm very excited to see that the DNN-based algorithm is able to overcome the influence of 3-D effects (illuminating and shadowing) and yield retrievals in close agreement on LES. My biggest concern is that if this result is robust enough. I hope I am not mistaken, but it seems the DNN in this study is trained using only two LES cases in Figure 1 and moreover only applied to these two cases. If so, frankly, I not completely convinced if the algorithm will generate same successful retrievals if it is applied to other LES scenes or real satellite images. To convince me and the readers, the authors should consider a "blind test", in which they apply the algorithm to the LES cases other than the two training cases. For example, the authors can tweak the meteorological conditions in the LES (e.g., inversion strength, large-scale forcing etc.) to generate different cloud types/scenes, and then apply the DNN algorithm to assess and report its performance. r Overall, the authors need to demonstrate the robustness of their algorithm and results.*

**Response**: We understand this issue. Indeed, we limited our tests to two cases of boundary layer clouds and do not expect the DNN trained in this study perform well for cirrus cloud (for example) that is geometrically thick and optically thin. In general, NN is known to perform well for data that are similar to those used in the training. It is not very confident that the NN work well for data that are very different. This is a common issue in every NN/DNN-based cloud retrieval methods (Faure et al., 2011; 2012; Cornet et al., 2004; 2015; Evans et al., 2008; Kox et al., 2014; Minnis et al., 2014; Strandgen et al., 2017).

This study is just a feasibility study, but it is encouraging to see that the 3D radiative effects (e.g., illuminating and shadowing) are reasonably corrected in the results presented in this paper. The convolution filters shown in Fig. 8 "suggest" that the DNN indeed learned meaningful patterns of 3D radiative transfer although it is difficult to interpret how the filters correct the 3D effects. At least, in this study, we tried to expand the variety of training data by scaling the cloud optical thickness artificially. As a result, we could show the performance tests for a wide range of optical thickness from 0.1 to 100.

Additional simulations using a LES model is technically difficult for now because it is too expensive with time and computational cost in mind. More tests including different types of cloud should be done in the future works. In addition, when more cloud data are available, DNN should be retrained for additional cloud data. This will be important for practical applications in the future. We have added a few words in the last sentence in the conclusion as "...will require training using realistic cloud fields for various types of cloud."

*2) Complexity of the training: Note that 3-D radiative effects depend on many factors, not only just COT, CER and solar geometry, but also cloud top inhomogeneity, cloud geometrical thickness and surface reflectance among others [Várnai and Davies, 1999; Várnai and Marshak, 2001; 2002] as well as instrument characteristics. I'm wondering which ones of these factors have to be part of the training and which ones do not need to be. Take surface reflectance for example. Can we train the algorithm using only one surface reflectance and then it will work for all other types of surface? In addition to 3-D effects, the retrievals are also affected by many other factors, the presence of drizzle, atmospheric absorption, surface reflectance etc. It is not clear from the paper to what extent these factors are considered in the DNN algorithm training, and which ones are not. Overall, I'm trying to figure out how "smart" the algorithm is. If we have to worry about all the above-mentioned details in the training, then the practical usefulness of the algorithm becomes questionable.*

**Response**: In this study, we assumed surface reflectance of zero for simplicity to study a

feasibility of DNN-based retrieval methods. When analyzing actual satellite data, it would be possible to approximately correct the surface reflection component in observed radiance to get cloud-only radiance, if we know the surface reflectance. On the other hand, cloud-top roughness, geometrical thickness and vertical inhomogeneity within cloud are included in the training as they are from the LES cloud data. This is because we wanted to test whether the DNN can retrieve cloud column properties (or correct the 3D effects) even with the complexities that likely appear in real clouds. As the reviewer pointed out, the 3D radiative effects depend on many factors. Basically, all such important factors should be included in the training of DNN. This is a next step for practical applications to actual observation data. If we do not think deeply, we may just add solar geometry, surface reflectance and more parameters to the input vector of DNN. Although such a simple addition should be easy with the help by the current techniques, an appropriate DNN architecture for addition of input parameters should be investigated. We have added the last sentence in the conclusion as "An appropriate DNN architecture for addition of input parameters should be investigated in the future."

*3) Cloud mask: It is not clear from the paper how cloud masking is treated in the retrieval/training. If retrievals are done at the resolution coarser than the LES grid, then some pixels are inevitably partly cloudy. How are the partly cloudy pixels treated in the retrievals and training?*

**Response**: Subpixel clouds are not considered in this study. Horizontal resolution is 280 m for both training and test. We have added a sentence in the first paragraph in Section 2.1: "The area averaging was done over a cloud region of 280 m for x- and y-axis; For simplicity, subpixel clouds are not considered in this study."

*4) Definitions of CER: When cloud microphysics varies both vertically and horizontally, then the definition of CER can be very tricky. For example, Eq. (1) applies well to a single LES cell, no problem. (the root and meaning of the parameter need to be explained in detail though). The equation (2) for column-mean CER becomes tricky. First of all, does the vertical average takes into account any vertical weighting for example due to photon penetration depth [Platnick, 2000; Miller et al., 2016]? Some explanations are needed either way. Second, what the column-mean ? How to compute it? Third, what is the significance of the column-mean CER in Eq. (2)? Does it help understand the cloud radiative effects? Does it help the modelers validate their cloud microphysics simulations? Can it be used in combination of COT retrieval to estimate LWP? After defining the column-mean CER for a single column, the authors also need to explain how to aggregate/define the CER over multiple LES columns horizontally. For example, if the retrievals are done at 10x10 pixels, and each pixel has a slightly different column-mean CER, then what is the CER for the 10x10 pixel ensemble?*

*There are a few recent studies that discussed this topic. Maybe they are helpful [Miller et al., 2016] and [Alexandrov et al., 2012]*

**Response**: We thank the reviewer for this comment and suggestions of references. There is no consensus on a representative CDER definition for cloud column, in the community. We needed to define some representative CDER to treat vertical inhomogeneity in the retrieval. First we made coarse resolution (280 m) data of LWC and N. Then Eq. (1) and (2) are applied to define local CDER and column mean CDER. As in Eq. (2), the column mean CDER (Re) is without vertical weighting. Although we think the definition is enough for our current purpose, this issue will be focused when sub-pixel cloud inhomogeneity is in mind. We have added a few sentences in Section 2.1:

"There is no community consensus on a single definition of CDER that is representative of the full column in the case of a vertically inhomogeneous cloud. Nevertheless, this study introduces the retrieval of such a representative CDER, …"

"It should be pointed out that there are other possibilities for column-average CDER (Miller et al., 2016). "

*5) Plane-parallel albedo bias: This study focuses on the impacts caused by IPA, but there is another type of bias, plane-parallel-albedo bias (PPHB). It is not clear to me if the DNN described in this study could also take care of the PPHB. Note that recently, Zhang et al. [Zhang et al., 2016] described a novel method to correct the PPHB, which might be helpful for this study.*

**Response**: In principle, the DNN should be able to handle PPHB as is shown by pioneering work using NN in Faure et al. (2001). We have added a citation to Zhang et al. (2016) in the Introduction, as follows:

"Zhang et al. (2016) recently described a novel method to correct the effect of in-pixel cloud inhomogeneity using subpixel reflectance variabilities."

*6) Lack of technique details: I agree with the other reviewer that many important technique details are lacking from the current paper. Currently, the paper is rather short, so there is plenty of space to add in more detailed description and discussion, especially for Section 3 Method. Just to give an example, what are the meaning of Eq. (8) and (9)? Why do they provide the "relationships between inputs and outputs variables" of DDN, what kind of relationship?*

**Response**: According to a comment from the referee 1, we have moved explanations on fundamental DNN techniques to Section 3.1, where fundamental deep learning techniques are summarized. We have added explanations in several parts. We hope revised manuscript is easier

to read. However, we have not added very long explanations on the techniques that are really technical and not essential to the conclusions of this paper. We think it is better to leave the technical details of each optimization and deep learning technique for readers to consult the textbooks or references cited in the current manuscript. The deep learning techniques are rapidly growing in broad areas, upon many successes in engineering and applications for the artificial intelligence. Essential characteristics of each deep learning techniques are described in the current manuscript.

Although DNN can generally approximate nonlinear functions, it is expected that DNN may approximate the functions with a smaller number of DNN layers if the nonlinearity is less. The functions in Eq. (8) and (9) are less nonlinear to radiances. This kind of simple transform help better performance of DNN retrievals. We have modified the introductory sentences as follows: "Although DNN can generally approximate nonlinear functions, it is expected that less nonlinear functions can be approximated by fewer DNN layers. Constructing efficient DNN thus makes it desirable to linearize the relationship between input and output variables to some degree. Because the radiances are highly nonlinear with respect to the COT and CDER, it is convenient to transform the COT and CDER by some simple functions."

Reply to anonymous referee 2

First of all, we would like to thank the reviewer for reading our paper carefully and providing constructive comments. In the revised manuscript, we have tried to accommodate all the suggested changes. The modifications from the originally submitted version are highlighted in the revised manuscript. Please see our specific responses below.

*The paper describes a new technique for satellite measurements of cloud optical thick- ness and cloud droplet effective radius. The key feature of the technique is that it takes into account 3D radiative effects and subpixel variability by considering not one pixel at a time, but by performing simultaneous retrievals over 10 by 10 pixel areas. The most important aspect of the technique is the use of a deep learning algorithm. This is a significant new development, and the study makes an important contribution on the path toward more accurate satellite retrievals of cloud properties. Overall, the method- ology is sound and the presentation is suitable. However, I believe that a few important improvements are needed in the analysis. My recommendation is therefore to make some major revisions. Please find below my detailed comments.*

*comments*

*Major issues:*
*1.*
*Page 7, Line 8 mentions that "The test dataset used for evaluation should be indepen- dent of the training dataset." My sense is that in this initial study the training and testing datasets are not fully independent, as they come from the very same cloud fields, and that this would be good to mention. (The two datasets include different randomly se- lected locations within the cloud fields, but the statistics of cloud properties are identical in the training and testing datasets.)*
*As noted in Page 10, Lines 7-8, it will be an important future step to examine the per- formance of the retrieval for a wider range of cloud parameters. It is reasonable to leave this (and the evaluation based on fully independent training and testing datasets) to a future paper, but even the current results could offer further insights into the ro- bustness of the proposed retrieval algorithms. Most importantly, one could examine not only the overall results, but also separately the results for open-cell and closed-cell convection cases. This would demonstrate that the same algorithm and training set improves retrieval accuracy for two very different types of cloud structures. I don't think the currently presented results show this: Overall error statistics may conceivably im- prove due to improvements for open-cell convection only, without any improvements for closed-cell convection. (Because retrieval uncertainties are likely larger for open-cell convection, it may be best to examine by what percentage DNN-2r and DNN-4w reduce the retrieval errors of IPA retrievals for open-cell and for closed-cell convection.) The paper did a good job in examining results as a function of optical thickness, but the new*

*analysis of already performed retrievals would help because open-cell and closed-cell convection cases differ in horizontal structure even at locations where vertical optical thicknesses are similar.*

**Response**: We have added Table 2 to show RMS errors (in %) for open and closed cell cases, separately. The results show that DNNs generally retrieve more accurate COT and CDER than IPA method. For SZA of 60 deg., DNNs are better than IPA in both open and closed cloud cases, while there is an exception; IPA is better for COT for SZA of 20 deg. in the closed cell case. We have added explanations about this table, as follows:

"Table 2 shows the relative RMSE of estimated COT and CDER by the IPA and DNN retrievals for open-cell and closed-cell cases. In both cases, the retrieval accuracies for DNNs are obviously improved compared to the IPA retrieval. An exception appears for COT in closed-cell case when SZA is 20°; COT RMSE of 24% for DNN is larger than 16% for IPA. The DNN-4w is better than DNN-2r in both cases."

As the reviewer suggested, we have revised the explanation about test datasets as "the training and test dataset include different randomly selected locations within the cloud fields, but the statistics of cloud properties are identical in the training and test datasets."

*2.*

*Page 5, Lines 13 and 22: I wonder why scene parameters are estimated for 8 X 8 pixel arrays when using the DNN-2r method, but only for the central 6 X 6 pixel arrays when using the DNN-4w method. This could make sense if 3D effects acted over larger distances at 3.75 microns than at the 0.86 and 2.15 microns used by the DNN- 2r method, but neither my own physical reasoning nor the filter weights in Figure 8 suggest this. In fact, Figure 8 shows that DNN-4w retrievals at a pixel are strongly affected by 0.86 micron radiances 2 pixels away. This suggests that (at least for pixels at the edges of 8 X 8 pixel areas) the DNN-2r method cannot capture the portion of 3D effects caused by areas more than a pixel away. This probably contributes to DNN- 2r giving less accurate results than DNN-4w (a tendency mentioned in Page 9, Lines 31-32) and should be mentioned in the discussion of the differences between the two methods at the top of Page 10. (The discussion should also include the impact of additional wavelengths in DNN-4w and algorithmic differences.) Also, it could help to clarify explicitly in the paper whether DNN-2r retrievals inside (not along the edges of) 8 X 8 pixel areas are affected by radiances 2 pixels or more away. If they were, it could even make sense to analyze retrieval accuracy only for pixels in the central 6 X 6 pixels of 10 X 10 pixel areas (similarly to DNN-4w).*

**Response**: We appreciate this comment. Honestly speaking, we had no strong reason to limit the output pixels in DNN-2r and set it as 8x8 pixels. On the other hand, because of convolution filters of 5x5 pixel size, the number of pixels were limited as 6x6 pixels. As the reviewer

pointed out, the difference of the number of pixels in the DNN output layer may have an effect on the retrieval performances. However, we think that it is not a main reason because 1) influences from neighboring pixels generally tend to weaken with increasing distance from the target pixel, 2) there are only 28 edge pixels in the 64 (8x8) pixels area, and 3) most of these edge pixels just miss only one side of neighboring pixels 2 pixels away. The 3D radiative effects actually operate on larger horizontal scales than 2 pixels (560 m) as captured by DNN-4w, while the effects tend to weaken for larger scales. As shown in Fig. 8, the filter coefficients do not vanish at the edges of the window. To improve retrieval accuracy, it is better to increase the number of adjacent pixels used for both DNN-2r and DNN-4w, as this point is mentioned in the conclusion.

In the initial stage of this study, we tested various kinds of DNN architectures with a different number of wavelengths, the use of convolutional layer, different activation function, and so on. For example, we tested one similar to DNN-2r but using four wavelengths or one similar to DNN-4w but using only two wavelengths. However, the DNN-2r and DNN-4w were the best performed ones. Therefore, at least we can say that a main reason for the better retrieval performance for DNN-4w than DNN-2r would be a combined effect of use of additional wavelengths and a different DNN architecture (e.g., use of convolutional layer and release from the reliance on the IPA retrieval). However, unfortunately, we could not make the reason very clear. We have added the description regarding the future improvements in the manuscript, as follows:
"The above two DNN structures were obtained from various trial-and-error experiments. Different DNN structures were also tested. For example, we tested a DNN similar to DNN-2r but with four wavelengths, and one similar to DNN-4w but with only two wavelengths. However, DNN-2r and DNN-4w performed best. There is room for improvement in DNN structures, which should be investigated in the future."

*Minor issues:*

*Page 1, Line 23: What is meant by "cloud state"?*

**Response**: In this context, "cloud state" means cloud vertical and horizontal inhomogeneity. We replaced these words in the new manuscript as "cloud horizontal and vertical inhomogeneity".

*Page 2, Line 23: The study by Evans et al. ("The Potential for Improved Boundary Layer Cloud Optical Depth Retrievals from the Multiple Directions of MISR", J. Atmos. Sci., 2008) should also be mentioned, as it also used a neural net for cloud retrievals.*

**Response**: We have added Evans et al. (2008) as a reference.

*Page 2, Lines 26-27: What is meant by "feature" and "feature extraction"?*

**Response**: In this context, "feature" means a feature in the training datasets. This word is usually used in the classification problem (e.g., object recognition from input image) using DNN. We have modified the manuscript as follows:
"features in training datasets are learned hierarchically in the DNNs, although it is not very easy know how the features are described in the DNNs"

*Figure 2: It would help to indicate the time elapsed during the 60 time steps along the horizontal axis, or to mention the time step in the figure caption.*

**Response**: We have added an explanation in the Fig. 2 caption as
"A time step corresponds to one minute."

*Figure 3: It would help to clarify why there is a fully connected layer near the top of the left column that operates only on radiances and not on the IPA-estimated scene parameters.*

**Response**: We thought the layer should be in, at least, an either side (left or right side) of the figure. A layer can be inserted in both sides, but we did not have very strong reason. We believe this is a minor issue, but we have added a sentence as follows:
"In the first layers, radiances and IPA-estimated cloud properties are merged to obtain 8x8x2 elements (two elements per pixel for 8x8 pixels)."

*Page 6, Lines 27-29: It would help to clarify whether all pixels within an LES scene are multiplied by the same randomly chosen number, or all individual pixels are multiplied by a different number. (My guess is the first option.)*

**Response**: As the reviewer guessed, all pixels within a single scene are multiplied by the same number. We have added a sentence:
"The cloud extinction coefficients of all pixels within a single cloud scene are multiplied by the same number."

*Page 6, Line 25 and Page 7, Lines 8-10: What does the word "samples" refer to? My guess is that each sample is a 10 X 10 pixel area. If my guess is right, can samples overlap? Also, it would help to mention the total number of pixels in the LES dataset, as this could show whether the training set includes almost all LES pixels or just a small fraction of them.*

**Response**: As the reviewer guessed, a "sample" here means a 10 x 10 pixel area, and sample areas can be overlapped by other samples. The horizontal size of a LES scene is 28km x 28km. The total number of pixels in one LES scene is 100 x 100 = 10000 pixels. We have added some descriptions as mentioned above.

*Figure 7: It could help to include into one of the panels a PDF of true optical thickness values.*

**Response**: Figure 1 shows means and standard deviations of COT for the two cloud cases, from which rough shapes of COT PDF may be imagined. Although the distribution functions of true COT are not presented in this paper, the following figures show the joint histograms of the DNN-4w retrieval error and COT.

[Figure]

*Figure 9: The legend should indicate which color shading corresponds to which line/method.*

**Response**: We have revised Figure 9 as suggested.

*Page 10, Lines 5-6: I am not sure the sentence "In the DNN-4w that we tested, we excluded 3D radiative transfer effects that occurred at horizontal scales greater than approximately 1.5 km (5 pixels)" is correct. Based on Figure 8, I thought that DNN-4w retrievals exclude 3D effects that occur at horizontal scales greater than 2 pixels (560 m). This is because I thought the pixel*

*whose properties we are retrieving is at the center of the filters in Figure 8, which means that only radiances two pixels away are considered. A correction of this sentence or a clarification of the meaning of filters in Figure 8 would help.*

**Response**: The reviewer is right. We have fixed the sentence as
"...scales greater than 560 m (2 pixels)"

*Somewhere in the text it would help to comment on whether the speed of calculations would be a concern for using DNN in operational retrievals in the near future. (For example, how does the speed of DNN compare to the speed of IPA and NN retrievals?)*

**Response**: We did not exactly compare the computational costs of DNN, IPA, and NN retrieval. We have added an explanation about rather general facts about the DNN computational cost, in the last part of Section 3.3, as follows:

[revised manuscript text omitted]